# Asymmetry in Galaxy Spin Directions—Analysis of Data from DES and Comparison to Four Other Sky Surveys

Lior Shamir

Department of Computer Science, Kansas State University, Manhattan, KS 66506, USA; lshamir@mtu.edu

**Abstract:** The paper shows an analysis of the large-scale distribution of galaxy spin directions of 739,286 galaxies imaged by DES. The distribution of the spin directions of the galaxies exhibits a large-scale dipole axis. Comparison of the location of the dipole axis to a similar analysis with data from SDSS, Pan-STARRS, and DESI Legacy Survey shows that all sky surveys exhibit dipole axes within 52° or less from each other, well within $1\sigma$ error, while non-random distribution is unexpected, the findings are consistent across all sky surveys, regardless of the telescope or whether the data were annotated manually or automatically. Possible errors that can lead to the observation are discussed. The paper also discusses previous studies showing opposite conclusions and analyzes the decisions that led to these results. Although the observation is provocative, and further research will be required, the existing evidence justifies considering the contention that galaxy spin directions as observed from Earth are not necessarily randomly distributed. Possible explanations can be related to mature cosmological theories, but also to the internal structure of galaxies.

**Keywords:** galaxies; large-scale structure; cosmic anisotropy





## 1. Introduction

The advancement of research instruments and information systems have enabled new types of data-driven research that was not practical in the pre-information era. These observations include very large structures [1–5] that could be beyond astrophysical scale, and therefore challenging the cosmological principle. Other observations include probes that show cosmological-scale anisotropy or other anomalies [6]. Such probes include the cosmic microwave background [7–24], radio sources [25–30], short gamma ray bursts [31], LX-T scaling [32], cosmological acceleration rates [33–35], Ia supernova [36,37], galaxy morphology types [38], dark energy [33,39–41], fine structure constant [42], galaxy motion [43], $H_0$ [44], polarization of quasars [45–49], and cosmic rays [50–54]. It has also been shown that the large-scale distribution of galaxies in the Universe is not random [55].

These observations are not necessarily aligned with the standard cosmological models [6,33,35,44,56–60], and can require certain expansions of the current models. Possible explanations that can fit these observations include double inflation [61], contraction prior to inflation [62], primordial anisotropic vacuum pressure [63], multiple vacua [64], moving dark energy [65], and spinor-driven inflation [66]. Other proposed theories are ellipsoidal universe [10,67–70], geometric inflation [71–74], flat space cosmology [75–78], supersymmetric flows [79], and rotating universe [80], while early rotating universe theories assumed a non-expanding universe [80], more recent models are based on rotation with cosmological expansion [81–86]. In these models, a cosmological-scale axis is expected.

The existence of a cosmological-scale axis can also be associated with the theory of black hole cosmology, which can explain cosmic accelerated inflation without the need to assume the existence of dark energy [87–90]. Stellar black holes spin [91–96], and their spin is inherited from the spin of the star from which the black hole was created [94]. It has been therefore proposed that a universe hosted in a stellar black hole should have an axis and a preferred direction oriented around it [85,97–101]. Black hole cosmology is also linked to

theory of holographic universe [102–107], which can represent the large-scale structure of the Universe in a hierarchical manner [108,109]. Additional discussion about black hole cosmology is available in Section 6.

This study aims at analyzing possible cosmological-scale anisotropy using the probe of the large-scale distribution of spiral galaxies. The initial angular momentum of galaxies is largely believed to be driven by subtle misalignment between the tidal shear tensor and the Lagrangian patch, a theory known as *tidal torque theory* [110–119]. Tidal torque theory makes a direct connection between galaxy spin and the cosmic initial conditions, leading to a link between galaxy spin and the large-scale structure [117–120].

That contention is in agreement with numerous observations that show a link between the alignment of galaxy rotation and the filaments, clusters, and walls of the large-scale structure [121–134]. The alignment in the spin directions has been also observed with galaxies that are too far from each other to have any kind of gravitational interactions [135,136], and a statistically significant correlation was observed between the direction of rotation of galaxies and the initial conditions [134]. Numerical simulations also showed a connection between galaxy rotation alignment and the large-scale structure [133,137–143].

Another possible agent that can determine galaxy spin is galaxy mergers [144,145], as well as dark matter halos that do not merge but pass close to each other [146]. Galaxy mergers are not directly dependent on the initial conditions, and therefore if galaxy mergers were the sole agent of galaxy angular momentum the distribution of galaxy spin would have been expected to be random [147]. Other observations showed that rotating galaxies were present in the early Universe, before their spin could be initiated by gravitational interactions [148,149].

Analysis of galaxies from the Sydney-Australian-Astronomical-Observatory Multi-object Integral-Field Spectrograph (SAMI) survey provided evidence of alignment of spins of galaxies with lower stellar mass with the parent structure, and perpendicular alignment compared to the parent structure when the galaxies are of higher mass [131]. That *spin-flip* phenomenon [131] and the dependency between the spin alignment and mass was also shown by numerical simulations [127,129,150–156] and theoretical models [157].

Evidence showing that the large-scale distribution of galaxy spin directions is not necessarily random have been also discussed in several previous studies covering large parts of the sky [135,136,158–167]. These observations include several different instruments such as SDSS [159,162,165], Pan-STARRS [162], HST [161], and DECam [166].

This paper is based on new data from the Dark Energy Survey (DES), and new analyses of data from several other sky surveys. The different sky surveys cover different parts of the sky, including both the Northern and Southern hemispheres.

## 2. Data

Data from five different sky surveys were used, including both Earth-based and space-based instruments. Earth-based sky surveys include the DESI Legacy Survey, the Dark Energy Survey (DES), the Panoramic Survey Telescope and Rapid Response System (Pan-STARRS), and the Sloan Digital Sky Survey (SDSS). Space-based data are taken from the five HST fields of the Cosmological Evolution Survey (COSMOS), the North Great Observatories Origins Deep Survey (GOODS-N), the South Great Observatories Origins Deep Survey (GOODS-S), the Ultra Deep Survey (UDS), and the Extended Groth Strip (EGS). Data from the Earth-based sky surveys were annotated automatically, while HST data was annotated manually through a labor-intensive process that also accounted for possible human perceptual bias as will be described in Section 2.6. As will be discussed in Section 5, a smaller dataset of manually annotated SDSS galaxies was also used.

Due to the size of the data, the image format used for the ground-based sky surveys is the JPEG format. The only exception is HST, where the far smaller size of the data allowed downloading the images in the FITS format. The analysis of the data requires downloading the images to a local server, where the images can be annotated. That was done by using the *cutout* service of the respective sky surveys. Because the FITS is an image format that is

not necessarily efficient in terms of file size, downloading such a high number of images in the FITS format becomes impractical. For instance, downloading the DES images in the JPEG format lasted over six months of continuous data downloading. The file size of a typical single JPEG image is about 20 KB. The file size of a similar image in the FITS format is more than 300 KB for each channel. With the same bandwidth used to download the JPEG images, downloading the same images in three channels using the FITS format would have required more than 20 years.

Unlike the FITS files, images in the JPEG format do not allow reliable photometry, but they provide visual information about the morphology of the object. The visual information is the information used in this study for the morphological analysis of the galaxies. The JPEG images were converted to grayscale [168], before being annotated by their spin direction as will be described later in this section. By converting the images to grayscale, each image was annotated once, rather than several different times for each color channel. The analysis of the spin directions is done by the distribution of intensities of the pixels to identify the galaxy arms, as will be described in Section 2.1. The JPEG format might not provide accurate photometry as the FITS file format, but it provides sufficient information to identify the arms of the galaxies, and consequently their spin direction. The use of the JPEG format instead of the FITS format might lead to a higher number of galaxies that whose spin direction cannot be determined. Examples of cases of galaxies with clear spin directions that cannot be identified from the images are provided in Section 4.9. That, however, is expected to affect images of galaxies that spin clockwise in the same manner as it affects images of galaxies that spin counterclockwise.

### 2.1. Automatic Annotation of Galaxies by Their Spin Direction

Modern astronomical sky surveys such as DES collect a very large number of galaxy images, making it impractical to annotate them manually. Automatic annotation of the galaxies also has the advantage of being insensitive to human perceptual bias. Obviously, when using pattern recognition or deep learning systems that are based on manually annotated training data, human perceptual bias might still impact the analysis. However, when using model-driven approaches, with no training data and no manual intervention, no human bias is expected. Moreover, such a model can be designed in a manner that makes it mathematically symmetric.

While the use of machine learning, pattern recognition, and specifically deep learning has been becoming highly popular for the purpose of automatic image annotation, these methods are based on complex data-driven non-intuitive rules. Deep neural networks tend to provide different results based on the number of training images or even the order by which the training images are used. The complexity of their rules leads deep neural networks systems to situations in which they make "accurate" predictions even when the image data has no interpretable information [169,170], showing that these systems learn from contextual information [169] or even from seemingly blank background [170] rather than the image analysis problems they intend to solve. Such bias of deep neural networks has been shown specifically in the analysis of galaxy images [171]. Additionally, training such pattern recognition systems requires a manually annotated training set. Such training data could be subjected to the perceptual bias of the humans who annotated the images. A possible solution is to use unsupervised machine learning, which is not necessarily subjected to the preparation of a manually annotated dataset that is used for training and testing the algorithm. However, since unsupervised machine learning cannot be controlled by defined formal rules, it is also difficult to ensure their symmetry. Since galaxies are very different from each other, unsupervised machine learning can be affected by an unknown and uncontrolled number of factors that the algorithm identifies (e.g., size, shape, brightness) but are not related to the curve of the arms, while such algorithm can be used, their results are more difficult to validate compared to algorithms that use simple "mechanical" rules. Therefore, pattern recognition, and specifically deep neural networks, are imperfect for the task of identifying subtle asymmetries in the large-scale structure [171].

It has been shown with theoretical and empirical experiments that even a small bias in the annotation algorithm can lead to substantially biased results [165].

To perform a fully symmetric annotation, the Ganalyzer algorithm was used [168]. Ganalyzer is a model-driven algorithm that uses mathematically defined and intuitive rules that are also fully symmetric. It does not rely on complex data-driven rules, and does not rely on training data. While model-driven algorithms might also be subjected to bias, their defined rules make them easier to interpret, test, and design them in a symmetric manner. That reduces the possibility of an unexpected bias in the algorithm. Additional discussion about possible biases in the annotation algorithm is given in Section 4.1.

First, Ganalyzer transforms each galaxy image into its radial intensity plot transformation. The radial intensity plot of an image is a $35 \times 360$ image, such that the pixel $(x, y)$ in the radial intensity plot is the median value of the $5 \times 5$ pixels around coordinates $(O_x + \sin(\theta) \cdot r, O_y - \cos(\theta) \cdot r)$ in the original galaxy image, where $r$ is the radial distance measured in percentage of the galaxy radius, $\theta$ is the polar angle measured in degrees, and $(O_x, O_y)$ are the pixel coordinates of the galaxy center. The radial distance is between 40% and 75% of the radius of the galaxy. That avoids parts of the arms that are closer to the galaxy center, as well as parts of the arms that are in the outskirts of the galaxy, where the arms start to fade.

Arm pixels are expected to be brighter than non-arm pixels at the same radial distance from the center of the galaxy. Therefore, peaks in the radial intensity plot are expected to correspond to pixels on the arms of the galaxy at different radial distances from the center. To identify the arms, peak detection [172] is applied to the lines in the radial intensity plot, and then a linear regression is applied to the peaks in adjunct lines. The slope of the line formed by the peaks reflects the curves of the arm. Since backward winding galaxy arms are rare, the curve of the arm reflects the spin direction of the galaxy. Backward-winding galaxies are expected to exist, but these cases are rare, and are expected to be distributed equally among clockwise and counterclockwise galaxies. In the case of irregular galaxies the location of the center of the galaxy might not be always clear. However, most irregular galaxies do not have clear spiral arms. If the galaxy has clear arms but not a clear center the algorithm might fail to identify the correct spin direction. That, however, is not a common case, and these cases are expected to be distributed evenly between galaxies that spin clockwise and galaxies that spin counterclockwise. As explained in Section 4, such cases can reduce the magnitude and statistical significance of the asymmetry, but cannot artificially increase it. A more detailed discussion about backward winding galaxies and other effects that can affect the results is provided in Section 4.

Figure 1 shows examples of four DES galaxies. These galaxies can be found at coordinates $(\alpha = 0.3822°, \delta = -4.9716°), (\alpha = 0.9689°, \delta = -4.955°), (\alpha = 1.619°, \delta = -4.9877°)$, and $(\alpha = 1.7177°, \delta = -4.9853°)$. The figure also shows the radial intensity plot of each galaxy. Below each radial intensity plot the figure displays the peaks detected in the radial intensity plot after applying the peak detection algorithm [172]. As the figure shows, the alignment of the peaks in a certain direction reflects the winding of the arms, and therefore the sign of the linear regression deduced from the peaks reflects the direction towards which the galaxy spins. More information about Ganalyzer can be found in [162,166,168,173].

Elliptical galaxies might also have some peaks in their radial intensity plots. In that case, the peaks are expected to form a straight line, and with no preferred direction. An example can be found in Figures 2 and 3 in [168], showing the radial intensity plot and the peaks, respectively.

Obviously, not all galaxies are spiral galaxies, and not all spiral galaxies have a clearly identifiable spin direction. It is therefore clear that most of the galaxies cannot be used for the analysis due to the inability to identify the direction in which they spin. For that reason, only galaxies that have 30 or more identified peaks in the radial intensity plot are used in the analysis. Galaxies that do not meet that threshold are not used regardless of the sign of the linear regression of their peaks. That is an important advantage of the

algorithm compared to common pattern recognition and supervised machine learning methods, in which an answer is forced even when the machine learning system cannot determine the case.

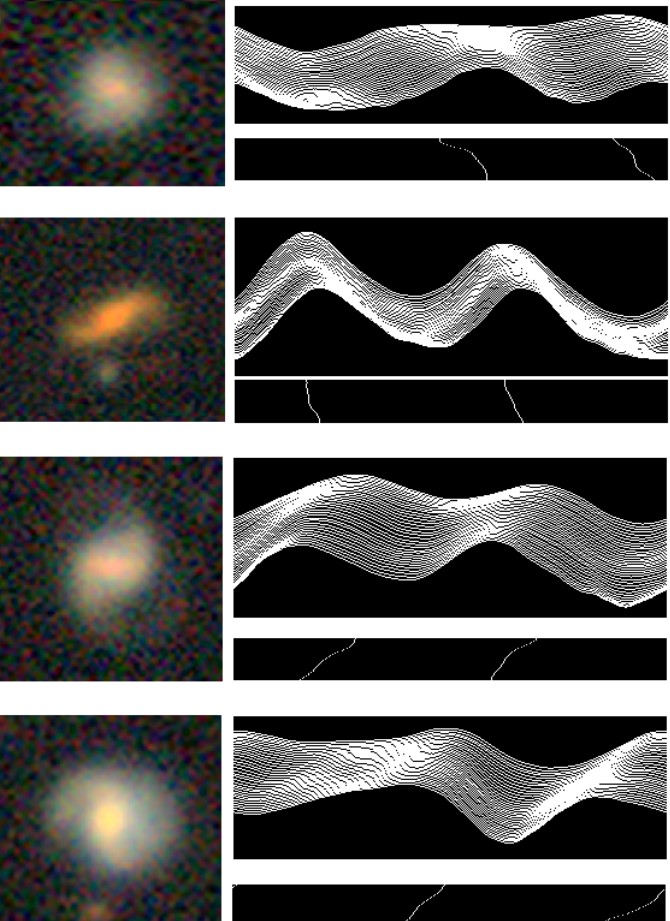

**Figure 1.** Examples of DES galaxy images and their corresponding radial intensity plots. The peaks detected in the radial intensity plots are displayed below the radial intensity plots. The X-axis is the polar angle of the pixel, and the Y-axis is the radial distance. The directions of the lines formed by the peaks reflect the change in the position of the arm compared to the galaxy center, and therefore reflect the curves of the arms. The direction towards which the arms are curved consequently reflects the spin direction of the galaxy. The algorithm is symmetric, and it is model-driven with intuitive rules. It is not driven by complex non-intuitive data-driven rules commonly used in pattern recognition systems.

### 2.2. Dark Energy Survey Data

The first dataset used in this study is galaxy images acquired by the Dark Energy Survey [174–177] DR1 [175]. DES uses the Dark Energy Camera (DECam) with the 4 m Blanco Telescope [178] in the Southern hemisphere to scan a total footprint of $\sim$5000 degrees$^2$. It acquires images in five bands—g, r, i, z, and y. The dark energy's primary mission is studying dark energy and dark matter. However, as a powerful sky survey with superb imaging capabilities, DES can be used for much broader tasks in astronomy [179].

The DES images were retrieved through the DESI Legacy Survey server, which provides access to data from several different sky surveys, including DES. The initial list of objects included all objects identified as exponential disks, de Vaucouleurs $r^{1/4}$ profiles or round exponential galaxies, and are brighter than 20.5 magnitudes in one or more of the g, r or z bands. That list contained an initial set of 18,869,713 objects. The galaxy images were downloaded using the *cutout* API of the DESI Legacy Survey. The size of each

image is 256 × 256, and retrieved in the JPEG format. The JPEG images are crated from the calibrated z, r, and g bands of DECam as the R, G, and B values of the RGB pixels.

Each image was scaled using the Petrosian radius to ensure that the object fits in the image. Since in DES the pixel scale is 0.263″/pixel [180], the scale is determined by $r \times 2 \times 0.262$, where $r$ is the radius of the galaxy. All images were used by the exact same computer to ensure full consistency of all images. While there is no apparent reason for differences in the analysis between computer systems, using the same computer system was done to avoid any kind of possible differences between different libraries or different operating systems. For instance, differences between two sets of galaxies that were downloaded by two different systems might leave the possibility that some unknown differences between the systems led to different results. Using the same physical machine completely eliminates the need to inspect differences between different computer systems. The process of downloading the images started on 25 April 2021, and ended about six months later on 1 November 2021. System updates were disabled during that time to avoid any possible automatic changes in the system, although such changes are not likely to impact the way images are being processed.

Once the image files were downloaded, they were annotated by their spin direction using the method described in Section 2.1. That provided a dataset of 773,068 galaxies with identifiable spin directions. That is merely 4% of the initial set of objects. An analysis of the symmetry in the selection of galaxies is given in Section 4.9. Some of these objects could be satellite galaxies or other photometric objects that are part of the same galaxy. To remove such objects, objects that had another object in the dataset within 0.01° or less were also removed from the dataset. That provided a dataset of 739,286 galaxies imaged by DES. The annotation of the galaxies lasted 73 days of operation using a single Intel Xeon processor. Then, the images were mirrored using *ImageMagick* and annotated again to allow repeating the experiments with mirrored images. The purpose of the mirroring of the images was to ensure that there is no systematic bias in the annotation. That practice is discussed in detail in Section 4.

To check the consistency of the annotations, 200 galaxies annotated as clockwise and 200 galaxies annotated as counterclockwise were selected randomly and inspected manually, as was done in previous experiments [162,166]. None of the galaxies annotated by the algorithm as spinning clockwise seemed to be visually spinning counterclockwise, and none of the galaxies annotated as spinning counterclockwise were visually spinning clockwise. Obviously, that test is a small scale, and it is practically impossible to test all galaxies in the dataset. However, the test suggests that the number of misclassified galaxies is expected to be small compared to the total size of the data. More importantly, because the algorithm is symmetric, misclassified galaxies are expected to be distributed evenly between the different spin directions, and therefore cannot lead to asymmetry as explained theoretically and empirically in Section 4.

To ensure that the process of galaxy annotation is consistent, all images were analyzed by the exact same algorithm, the exact same code, and the exact same computer. That ensured that the analysis cannot be impacted by different settings of different computers or different nodes. System updates were disabled during that time to disable any changes to the system, although such changes are not expected to lead to bias as discussed in Section 4. Although there is no known computer system fault that can lead to differences in the annotation, full consistency was ensured by using just one computer system with a single processor.

Figure 2 shows the distribution of the galaxy population in different RA ranges. The figure shows the distribution of the DES galaxies in 30° RA bins, but also the RA distribution of the galaxies imaged by the other three Earth-based sky surveys that will be discussed later in this section. Naturally, the distribution of the RA is not consistent across the different sky surveys. The downside of DES compared to the other sky surveys used in this study is that its footprint size is smaller, making it less effective in analyzing and comparing different parts of the sky.

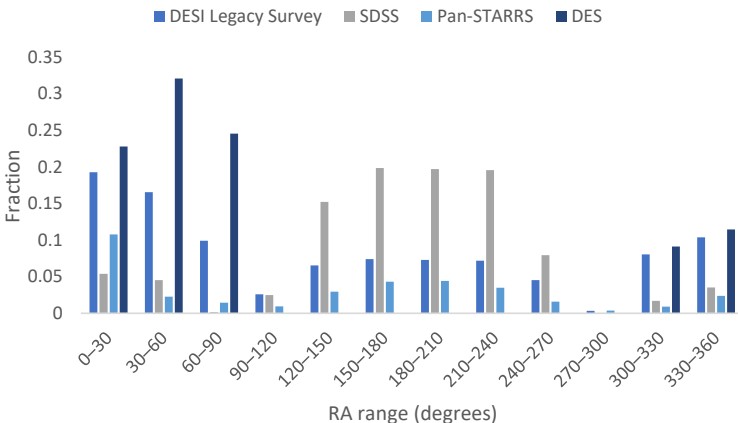

**Figure 2.** The distribution of the galaxies imaged by DES, DESI Legacy Survey, SDSS, and Pan-STARRS in different 30° RA ranges.

The distribution of the galaxies in different redshift ranges is shown in Figure 3. As with the RA, the figure shows the distribution of the redshift in all Earth-based sky surveys used in this study. Because the vast majority of the DES galaxies do not have spectra, the distribution of the redshift of these galaxies was determined by a subset of 12,290 galaxies that had redshift through the 2dF redshift survey [181].

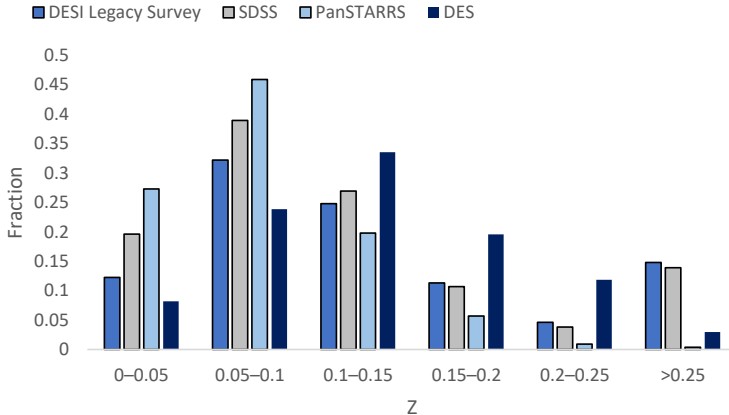

**Figure 3.** The distribution of the DES, DESI Legacy Survey, SDSS, and Pan-STARRS galaxies in different redshift ranges. With the exception of SDSS, the distribution was determined by a subset of objects with spectra.

## 2.3. DESI Legacy Survey Data

The DESI Legacy Survey [182] combines data collected by three different telescopes: the Dark Energy Camera (DECam) on the Blanco 4 m telescope, the Beijing-Arizona Sky Survey (BASS), and the Mayall z-band Legacy Survey (MzLS). The image acquisition is calibrated to provide a dataset of nearly uniform depth as described in [182]. The initial dataset contained 22,987,246 objects brighter than 19.5 magnitudes in their g, r, or z bands. The images were downloaded in a continuous process that required about six months, starting at the end of 2020 [166]. These images were acquired from the "south" bricks of Data Release 8 of the DESI Legacy Survey, and therefore did not include the entire footprint of the DESI Legacy Survey. The Northern sky was covered by SDSS and Pan-STARRS as described later in this section. The south bricks of the DESI Legacy Survey were acquired by using the DECam camera in the Blanco 4 m telescope. As with DES, the images were downloaded in the JPEG format, where the values of the RGB pixels are the calibrated z, r, and g bands of DECam.

The images were annotated to clockwise and counterclockwise galaxies in the same way the DES galaxies were annotated, and described in Section 2.1 or in [166]. After the annotation ended, the final dataset contained 807,898 galaxies. The dataset and the way it was acquired and annotated are described in detail in [166]. Because the DESI Legacy Survey and DES have overlapping footprints, 101,786 galaxies in the dataset also exist in the DES dataset described in Section 2.2.

The vast majority of the galaxies in the dataset do not have redshift. To statistically estimate the distribution of the redshift of the galaxies, 17,027 galaxies that had redshift through 2 dF [181] were used. Figure 3 shows the distribution of the galaxies in different redshift ranges. Figure 2 shows the RA distribution of the galaxies.

### 2.4. SDSS Data

SDSS [183] is one of the most impactful digital sky surveys, covering over $1.4 \times 10^4$ deg$^2$, mostly in the Northern hemisphere. The chosen subsample of SDSS galaxies has 63,693 galaxies [162]. The initial set of galaxies was retrieved from SDSS DR14, and included all galaxies that had spectra and their g-band Petrosian radius was 5.5″ or larger. These galaxies were annotated as described in Section 2.1, and the process of annotation provided 63,693 galaxies with identified spin directions. The dataset is described thoroughly in [162]. As with the DES images, the JPEG images were downloaded and used. A full description of the generation of the JPEG images from the FITS images of the different SDSS bands is described in [184].

The unique aspect of the SDSS dataset is that all galaxies have redshift. That allows normalizing the redshift distribution of the galaxies to match the redshift distribution of the other datasets. The preparation of that dataset is described in [162]. Figure 2 shows the distribution of the galaxies in different 30° RA ranges, and Figure 3 shows the redshift distribution.

### 2.5. Pan-STARRS Data

Another digital sky survey that was used is Pan-STARRS [185,186]. The initial Pan-STARRS set of galaxies included 2,394,452 Pan-STARRS objects with a Kron r magnitude was 19 or less, and identified as extended sources by all color bands [187]. As with the DES and SDSS images, the images were downloaded in the color JPEG format. These images are RGB images such that the R-value is the y band, the G value is the i band, and the B value is the g band.

These images were classified automatically by Ganalyzer [168] as described in Section 2.1, and with more details in [162,166,168]. That process provided 33,028 galaxies imaged by Pan-STARRS, and annotated by their spin direction. The distribution of the galaxies by their RA is shown in Figure 2. More information about the collection of the dataset and the distribution of the data can be found in [162]. The redshift distribution is shown in Figure 3, and was determined based on the distribution of 12,186 Pan-STARRS objects that had spectra in SDSS.

### 2.6. Hubble Space Telescope Data

Although there is no atmospheric effect that can make a galaxy that spins clockwise look as if it spins counterclockwise, space-based observation can eliminate the possible impact of such possible unknown and unexpected atmospheric effects. For that purpose, a dataset of space-based observations were prepared from the Hubble Space Telescope (HST) Cosmic Assembly Near-infrared Deep Extragalactic Legacy Survey [188,189]. The collection and preparation of that dataset are described in [161].

The dataset was taken from five different HST fields: the Cosmic Evolution Survey (COSMOS), the Great Observatories Origins Deep Survey North (GOODS-N), the Great Observatories Origins Deep Survey South (GOODS-S), the Ultra Deep Survey (UDS), and the Extended Groth Strip (EGS), providing an initial set of 114,529 galaxies [161].

The image of each galaxy was extracted by using *mSubimage* [190], and converted into $122 \times 122$ TIF (Tagged Image File) image.

Unlike the other sky surveys, the HST galaxies were annotated manually. Because the number of HST galaxies is smaller, a manual process of annotation becomes feasible. A manually annotated process can therefore lead to a dataset that is not just symmetric, but also complete, meaning that all galaxies that have a visually identifiable spin direction are included in it. The galaxies were annotated through a long labor-intensive process. During that process, a random half of the galaxies were mirrored for the first cycle of annotation, and then all galaxies were mirrored for the second cycle of annotation as described in [161] to offset the possible effect of perceptual bias. That provided a clean and complete dataset that is also not subject to atmospheric effects [161,166]. The total number of annotated galaxies in the dataset was 8690, and the distribution of the galaxies in the different fields is shown in Table 1.

**Table 1.** The number of galaxies in each of the five HST fields.

| Field Name | Field Center | # Galaxies | # Annotated Galaxies |
|---|---|---|---|
| COSMOS | 150.12°, 2.2° | 84,424 | 6081 |
| GOODS-N | 189.23°, 62.24° | 5931 | 769 |
| GOODS-S | 53.12°, −27.81° | 5024 | 540 |
| UDS | 214.82°, 52.82° | 14,245 | 616 |
| EGS | 34.41°, −5.2° | 4905 | 684 |

Due to the nature of the instrument, the galaxies imaged by HST are much more distant from Earth compared to the other telescopes, and their mean redshift is 0.58 [161]. Figure 4 shows the photometric redshift distribution of the HST galaxies, while the photometric redshift is highly inaccurate and systematically biased, it can provide a broad view of the manner in which the redshifts of the galaxies are distributed. As expected, the HST galaxies are far more distant from Earth compared to the galaxies imaged by the Earth-based sky surveys.

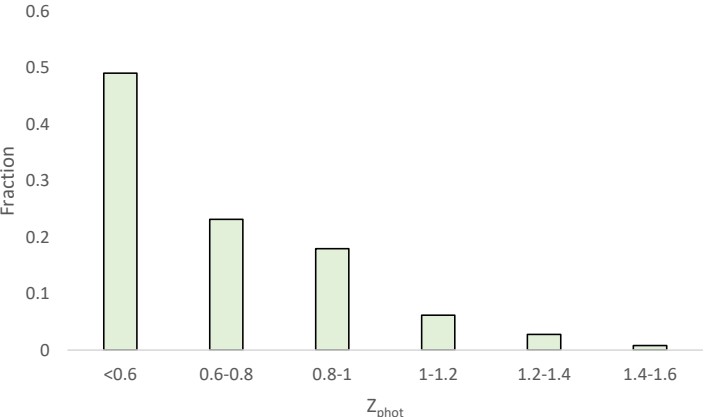

**Figure 4.** The redshift distribution of the HST galaxies based on the photometric redshift.

## 3. Results

The asymmetry $A$ in a certain field in the sky can be measured simply by $A = \frac{cw - ccw}{cw + ccw}$, where $cw$ is the number of galaxies spinning clockwise, and $ccw$ is the number of galaxies spinning counterclockwise. The standard error of the asymmetry can be determined by the normal distribution standard error of $\frac{1}{\sqrt{cw + ccw}}$. A simple analysis of the comparison between a possible asymmetry in different telescopes can be done by using HST COSMOS, and the other sky surveys that their footprint includes that part of the sky. The COSMOS field is centered at ($\alpha = 150.11°$, $\delta = 2.201°$). Obviously, that field is very small compared to

the other sky surveys, and therefore the field of ($145° < α < 155°$, $−3° < δ < 7°$) was used in DECam, SDSS, and Pan-STARRS. COSMOS is outside the footprint of DES, and therefore DES cannot be used in this analysis. All telescopes show the same direction of asymmetry.

As the table shows, all telescopes show a higher number of galaxies spinning clockwise around the field of COSMOS. These results can be compared to the distribution of the corresponding field in the opposite hemisphere, at ($325° < α < 335°$, $−7° < δ < 3°$). As Table 2 shows, in the same field in the opposite hemisphere all sky surveys show a higher number of galaxies that spin counterclockwise. The differences are not statistically significant, except for SDSS, but the table shows that the excessive number of galaxies spinning clockwise is not observed in the field in the opposite hemisphere that corresponds to the COSMOS field. The higher number of counterclockwise galaxies can indicate that the asymmetry observed around the COSMOS field is inverse when observing the same field in the opposite hemisphere, but the statistical significance might not be sufficient to make a strong conclusion regarding the existence of such inverse asymmetry. The absence of a higher number of galaxies spinning clockwise in that field indicates that the asymmetry observed in the COSMOS field is not necessarily driven by the bias of the image annotation algorithm, as such bias should have been observed in both fields. A more detailed analysis of these "sanity checks" is discussed in Section 4.

**Table 2.** The distribution of clockwise and counterclockwise galaxies in the field centered around the opposite hemisphere of the HST COSMOS field. The P values are computed by the accumulated binomial distribution assuming 0.5 mere chance probability of a galaxy spinning in a certain direction.

| Sky Survey | # Clockwise Galaxies | # Counterclockwise Galaxies | $p$ Value |
|---|---|---|---|
| Pan-STARRS | 97 | 117 | 0.09 |
| SDSS | 137 | 175 | 0.02 |
| DESI | 2450 | 2494 | 0.27 |

A galaxy that seems from an Earth-based observer to be spinning clockwise would seem to be spinning counterclockwise if the same galaxy was moved to the corresponding field in the opposite hemisphere. Therefore, if the distribution of galaxy spin direction is not symmetric, the asymmetry in one hemisphere is expected to be inverse to the asymmetry observed in the opposite hemisphere. Perhaps the most basic arbitrary separation of the sky into two hemisphere is one hemisphere is ($0° < α < 180°$), and the other hemisphere is ($180° < α < 360°$).

According to the cosmological principle, the Universe observed in the celestial Western hemisphere is expected to be the same as the Universe observed in the Eastern celestial hemisphere. The separation of the sky to the celestial Western and Eastern hemispheres is a simple separation that does not rely on any previous assumption or previous observation. Such separation allows applying simple statistical analysis, while it also ensures that the separation of the sky into two hemispheres is not "cherry-picked" for a certain specific condition, while the separation by the celestial coordinates does not have a specific cosmological meaning, it allows to use very simple statistical analysis, and the "blind" nature of the separation ensures that the analysis is not driven by the selection of specific parts of the sky.

Tables 3–6 show the number of galaxies in each hemisphere in each of the four Earth-based sky surveys. DESI, SDSS, and Pan-STARRS show inverse asymmetry in opposite hemispheres. As expected, the sign of the asymmetry in DESI is inverse to the sign of the asymmetry in SDSS and Pan-STARRS, as DESI is mostly the Southern hemisphere, while SDSS and Pan-STARRS cover mostly the Northern hemisphere. The asymmetry observed in Pan-STARRS agrees with SDSS, although the asymmetry is not statistically significant, possibly due to the lower number of galaxies in Pan-STARRS. When repeating the same analysis after mirroring the galaxy images, the results are inverse. That is expected due to the symmetric nature of the image annotation algorithm.

DES does not show opposite asymmetry in opposite hemispheres. That can be explained by the much more narrow RA range of the sky covered by DES, as shown in Figure 2. Figure 5 shows a comparison of the asymmetry in different RA ranges in the DES and DESI Legacy Survey. As the figure shows, in the hemisphere of (180°–360°), DES only has a population of galaxies in RA (300°–360°). In the ranges where the DESI Legacy Survey has a higher number of galaxies spinning counterclockwise, DES does not have any galaxy population, and therefore the difference in the asymmetry observed in the hemisphere (180°–360°) is expected.

**Table 3.** Distribution of clockwise and counterclockwise galaxies in opposite hemispheres in DESI Legacy Survey. The *p* values are the binomial distribution probability to have such difference or stronger by chance when assuming a 0.5 probability for a galaxy to spin clockwise or counterclockwise. Most galaxies in the DESI Legacy Survey used in this study are in the Southern hemisphere.

| Hemisphere | # *cw* Galaxies | # *ccw* Galaxies | $\frac{cw - ccw}{cw + ccw}$ | *p* |
|---|---|---|---|---|
| (0°–180°) | 252,478 | 250,555 | 0.0038 | 0.0033 |
| (180°–360°) | 151,948 | 152,917 | −0.0033 | 0.039 |

**Table 4.** Distribution of clockwise and counterclockwise galaxies in opposite hemispheres in SDSS. Most SDSS galaxies used in this study are in the Northern hemisphere.

| Hemisphere | # *cw* Galaxies | # *ccw* Galaxies | $\frac{cw - ccw}{cw + ccw}$ | *p* |
|---|---|---|---|---|
| (0°–180°) | 14,403 | 15,101 | −0.024 | 0.00002 |
| (180°–360) | 17,263 | 16,926 | 0.01 | 0.035 |

**Table 5.** Number of clockwise and counterclockwise galaxies in opposite hemispheres in Pan-STARRS.

| Hemisphere | # *cw* Galaxies | # *ccw* Galaxies | $\frac{cw - ccw}{cw + ccw}$ | *p* |
|---|---|---|---|---|
| (0°–180°) | 8725 | 8844 | −0.0067 | 0.18 |
| (180°–360°) | 7783 | 7676 | 0.0069 | 0.19 |

**Table 6.** Number of clockwise and counterclockwise galaxies in opposite hemispheres in DES.

| Hemisphere | # *cw* Galaxies | # *ccw* Galaxies | $\frac{cw - ccw}{cw + ccw}$ | *p* |
|---|---|---|---|---|
| (0°–180°) | 294,655 | 292,453 | 0.0038 | 0.002 |
| (180°–360°) | 76,327 | 75,851 | 0.0031 | 0.11 |

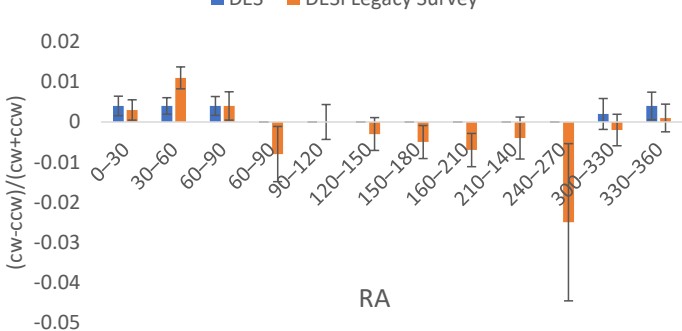

**Figure 5.** Asymmetry between clockwise and counterclockwise galaxies in the different 30° RA slices in DES and DESI Legacy Survey. DES does not provide data in the RA range of between 60° and 300°.

The separation of the sky into two hemispheres as shown in Tables 3–6 is an arbitrary separation of the sky, and was selected for the sake of simplifying the analysis, while the advantage of the analysis is its simplicity, it is also heavily dependent on the footprints of the sky surveys. To test whether the distribution of the spin directions of the galaxies exhibits a dipole axis, the galaxies in the datasets were fitted to cosine dependence from all possible integer $(\alpha, \delta)$ combinations. That was done by first assigning the galaxies with their spin direction $d$, which was 1 if the spin direction of the galaxy is clockwise, and $-1$ if the spin direction of the galaxy is counterclockwise.

Then, the cosines of the angular distances $\phi$ were $\chi^2$ fitted into $d \cdot |\cos(\phi)|$, where $d$ is the spin direction of the galaxy as was done in [159,161,162,166]. From each possible $(\alpha, \delta)$ combination in the sky, the angular distance $\phi_i$ between $(\alpha, \delta)$ and each galaxy $i$ in the dataset was computed. The $\chi^2$ from each $(\alpha, \delta)$ was determined by Equation (1)

$$\chi^2_{\alpha, \delta} = \Sigma_i \frac{(d_i \cdot |\cos(\phi_i)| - \cos(\phi_i))^2}{\cos(\phi_i)}, \tag{1}$$

where $d_i$ is the spin direction of galaxy $i$ such that $d_i$ is 1 if the galaxy $i$ spins clockwise, and $-1$ if the galaxy $i$ spins counterclockwise.

To determine the statistical significance of the possible axis at $(\alpha, \delta)$, the $\chi^2$ was also computed 1000 times, such that in each run the galaxies were assigned with random spin directions. Using the $\chi^2$ from 1000 runs, the mean and standard deviation of the $\chi^2$ when the spin directions are random was computed. Then, the $\sigma$ difference between the $\chi^2$ computed with the real spin directions and the mean $\chi^2$ computed with the random spin directions was used to determine the $\sigma$ of the $\chi^2$ fitness to occur by chance in that specific $(\alpha, \delta)$ combination [159,161,162,165,166]. Figure 6 shows the probabilities of a dipole axis in different $(\alpha, \delta)$ coordinates in SDSS, Pan-STARRS, DESI Legacy Survey, and DES.

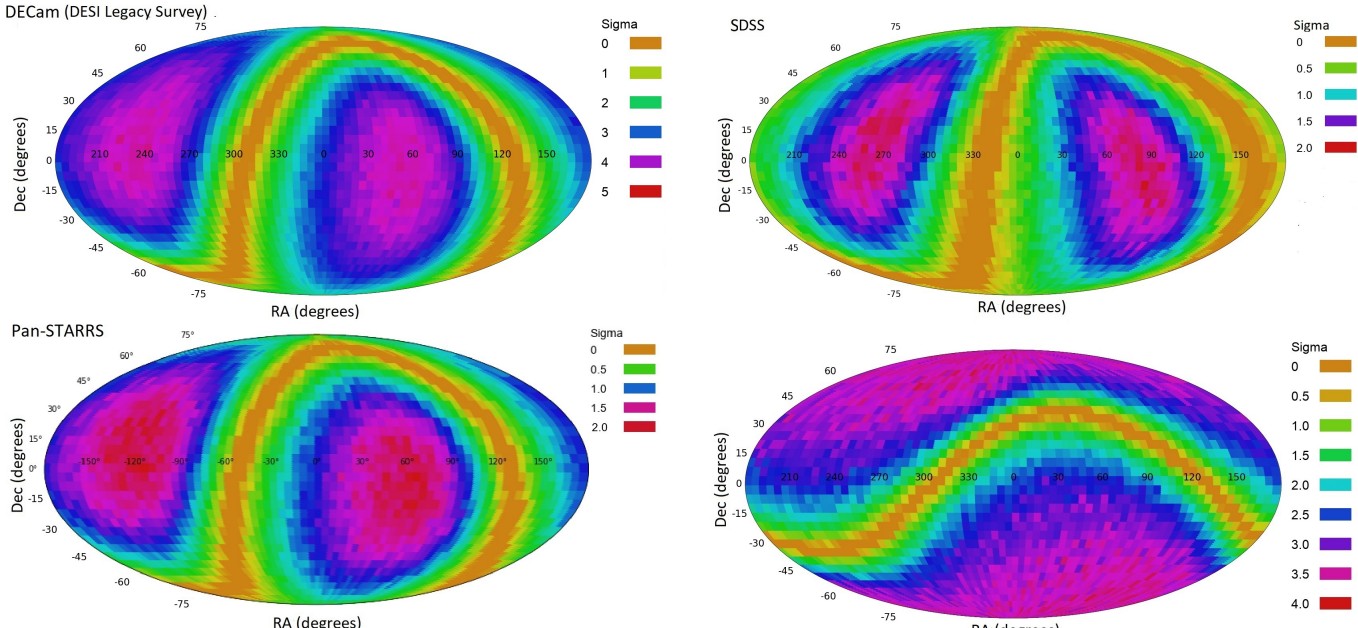

**Figure 6.** The statistical significance of a dipole axis in galaxy spin directions from different $(\alpha, \delta)$ combinations of SDSS, Pan-STARRS, DESI Legacy Survey, and DES. The SDSS galaxies are a subset selected such that the redshift distribution of the galaxies in that subset is similar to the distribution of the redshift in the DECam galaxies.

Previous results have shown that the location of the most likely dipole axis changes with the redshift, and datasets collected by different telescopes showed similar dipole axes when the distribution of the redshift in the datasets was similar [162,167]. The SDSS dataset

is unique compared to the other datasets in the sense that all galaxies have redshift. That allows normalizing the distribution of redshift in that dataset to fit the distribution of the redshift in the DESI Legacy Survey. As described in [167], the SDSS galaxies were selected such that their redshift distribution was similar to the distribution of the subset of DESI Legacy Survey galaxies with redshift obtained through 2dF. That resulted in a dataset of 38,264 galaxies such that the distribution of the redshifts of the galaxies fits the distribution of the redshift of the galaxies in the DESI Legacy Survey.

Table 7 shows the most likely axis observed in each of the digital sky surveys, as well as the $1\sigma$ error in the RA and declination. The table shows a certain agreement between the RA of the most likely dipole axes observed in the different datasets, all range between 47° (Pan-STARRS) and 78° (SDSS), and are well within the $1\sigma$ error range from each other. The agreement is despite the fact that each sky survey covers a different footprint and uses a different photometric pipeline, where DES has the smallest footprint of $\sim$5000 deg$^2$. It is also interesting that these axes are within $1\sigma$ error from the dipole axis formed by the $H_o$ anisotropy [34], the Planck CMB anisotropy dipole [24], the Australian dipole of variation of the fine structure constant [42], and it is also close to the CMB Cold Spot [191–195]. The Planck CMB anisotropy dipole reported in [24] is nearly identical to the dipole axis observed with the Pan-STARRS data, and well within $1\sigma$ from the dipole axes observed with the other telescopes.

**Table 7.** The most likely dipole axes observed in the different datasets, and the $1\sigma$ error range of the peak of the axes identified in each dataset.

| Dataset | RA (Degrees) | Dec (Degrees) | $\sigma$ | RA $1\sigma$ Error Range | Dec $1\sigma$ Error Range |
|---|---|---|---|---|---|
| DESI | 57 | −10 | 4.7 | 22–92 | −39–56 |
| SDSS | 78 | −12 | 2.2 | 27–124 | −67–61 |
| Pan-STARRS | 47 | −1 | 1.9 | 4–117 | −73–40 |
| DES | 75 | −47 | 3.7 | 332–123 | −90–16 |

When assigning the galaxies with random spin directions, the asymmetry disappears [159,161–163,165,166,196]. For instance, Figure 7 shows the probabilities of a dipole axis in different $(\alpha, \delta)$ combinations formed by the DES galaxies, when each galaxy is assigned a random spin direction. The most likely axis has a statistical signal of $0.67\sigma$.

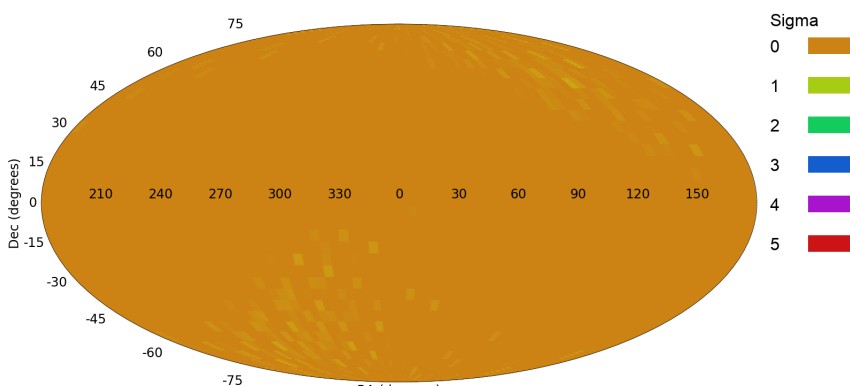

**Figure 7.** The statistical significance of a dipole axis in galaxy spin directions from different $(\alpha, \delta)$ combination in DES when the galaxies are assigned random spin directions.

Analysis was also done by combining galaxies imaged by several sky surveys into a single analysis. Such an experiment can increase the number of galaxies, but also the footprint size, and therefore can provide a more accurate location of the axis [196]. Combining the galaxies from SDSS, DESI Legacy Survey, Pan-STARRS, and HST datasets described in Section 2 led to a dataset of 958,841 galaxies [196]. Figure 8 displays the result of applying the analysis to the combined dataset. The analysis shows a dipole axis at ($\alpha = 47°$,

$\delta = -22°$), which is largely consistent with the axes of the specific sky surveys specified in Table 7. The statistical strength of the axis is $3.7\sigma$.

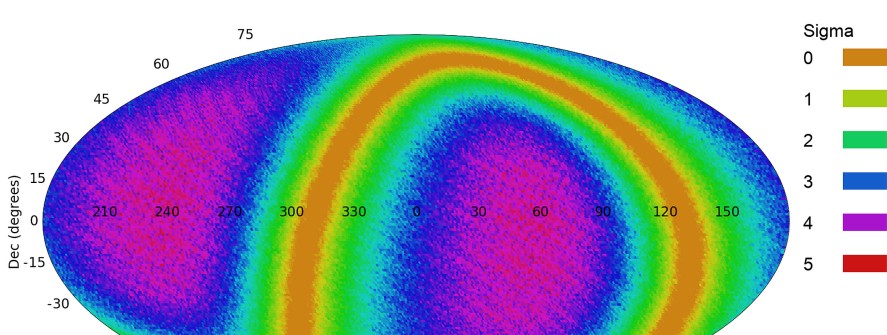

**Figure 8.** The statistical significance of a dipole axis from all integer $(\alpha, \delta)$ combinations in a dataset of 958,841 galaxies from SDSS, DESI Legacy Survey, Pan-STARRS, and HST.

## 4. Analysis of Possible Errors

Since the spin direction of a spiral galaxy is assumed to be merely the perception of the observer, the null hypothesis is that the spin directions of spiral galaxies are distributed randomly. Non-random distribution is therefore unexpected. One explanation for the observation would be an error in the analysis. This section discusses and explains several possible errors.

### 4.1. Error in the Galaxy Annotation Algorithm

A bias in the algorithm that annotates the galaxies by their spin direction can lead to asymmetry. However, several different observations indicate that the asymmetry in the distribution of spin directions of spiral galaxies cannot be the result of a bias in the galaxy annotation algorithm. The method used in this study to annotate the galaxies is a model-driven and fully symmetric algorithm that follows clear rules. It does not make use of machine learning or other complex rules driven by pattern recognition. Supervised machine learning and pattern recognition systems are often highly complex and unintuitive. Such systems are heavily dependent on the specific data they are trained with, and even seemingly meaningless implementation decisions such as the order of the training samples. An example of the symmetricity of the algorithm is shown in Figure 9, which shows the same analysis shown in Figure 5, but after mirroring the galaxy images. As expected, mirroring the galaxy images led to inverse asymmetry compared to the analysis with the original images.

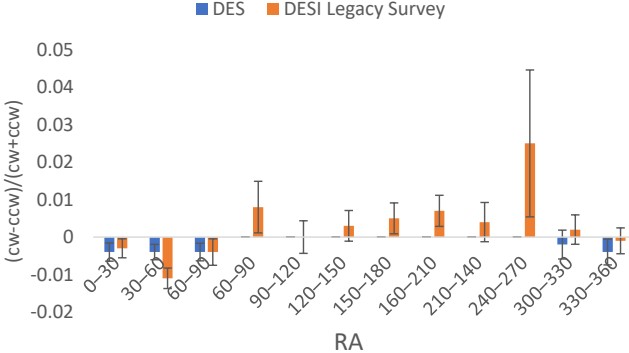

**Figure 9.** Asymmetry in different 30° RA slices such that the galaxy images are mirrored. The asymmetry is inverse to the asymmetry observed with the original images shown in Figure 5.

Another indication that shows that the observation is not the result of a bias in the algorithm that annotates the spin directions of the galaxies is that the observed asymmetry changes gradually between different directions of observations, and the sign of the asymmetry flips in inverse hemispheres. Because the galaxies are annotated independently, a bias in the algorithm that annotates the spin directions of the galaxies is expected to be consistent in different directions of observation. Certainly, it is not expected to show inverse asymmetry in opposite hemispheres. The retrieval of the image data and the annotation of the galaxy images were done on the same computer system, in order to avoid any kind of unknown and unexpected differences between the computers. While a computer program is naturally expected to provide the same results regardless of the machine it runs on, using the same computer system completely eliminates the possibility that any of the observations reported here are driven by differences between computer systems.

Since galaxies come in very different forms and shapes, it is possible that some galaxies, mainly irregular galaxies with unexpected shapes, can be annotated incorrectly. Since the algorithm is symmetric, any error that can exist in the algorithm is expected to have the same impact on galaxies spinning clockwise and galaxies spinning counterclockwise. If the algorithm that identifies the spin direction of the galaxies had a certain error, the asymmetry $A$ can be defined by Equation (2).

$$A = \frac{(N_{cw} + E_{cw}) - (N_{ccw} + E_{ccw})}{N_{cw} + E_{cw} + N_{ccw} + E_{ccw}},$$

(2)

where $E_{cw}$ is the number of clockwise galaxies incorrectly annotated as spinning counterclockwise, and $E_{ccw}$ is the number of counterclockwise galaxies incorrectly annotated as clockwise. When the annotation is symmetric, the number of galaxies spinning counterclockwise incorrectly annotated as galaxies spinning clockwise is expected to be similar to the number of clockwise galaxies misclassified as counterclockwise. Therefore, $E_{cw} \simeq E_{ccw}$ [165], and the asymmetry $A$ can be defined by Equation (3).

$$A = \frac{N_{cw} - N_{ccw}}{N_{cw} + E_{cw} + N_{ccw} + E_{ccw}}$$

(3)

Since $E_{cw}$ and $E_{ccw}$ must be positive numbers or zero, a higher number of galaxies annotated incorrectly makes the asymmetry $A$ lower. Therefore, a possible incorrect annotation of the galaxy images is not expected to show asymmetry in the data, and can only make the magnitude of the asymmetry lower.

As shown by an empirical experiment [165], when intentionally annotating some of the galaxies incorrectly, the results do not change substantially even when a high number of 25% of the galaxies are annotated incorrectly, as long as the error is applied evenly to galaxies that spin clockwise and galaxies that spin counterclockwise [165]. However, when the error in galaxy annotation is added in a manner that adds more incorrect annotations to a certain spin direction, even a mild error of just 2% leads to a strong asymmetry with extremely high statistical significance of over $10\sigma$. In that case, the dipole axis will peak exactly at the celestial pole [165].

It should also be mentioned that this study also uses a set of galaxies imaged by the Hubble Space Telescope that were annotated manually. The process of manual annotation was done by mirroring the galaxies to correct for a possible human bias. Another example of analysis with manually annotated galaxies will be shown in Section 5.

*4.2. Cosmic Variance*

It has been observed that galaxies as seen from Earth are not evenly distributed in the Universe. These small fluctuations in the density of the population of galaxies as seen from Earth are defined as "cosmic variance" [197,198]. Such variance can affect large-scale measurements at different parts of the sky and different directions of observation [199–201].

Asymmetry in galaxy spin direction is a relative measurement, making it different from other probes with absolute measurements such as the CMB. That is, asymmetry between galaxies with opposite spin directions is measured by using the difference between two different measurements made in the same part of the sky, which are the number of clockwise galaxies and the number of counterclockwise galaxies. Any cosmic variance that affects the number of clockwise galaxies that appear in a given field is expected to have the same impact on the number of counterclockwise galaxies in the same field.

### 4.3. Asymmetry in the Hardware or Software of Digital Sky Surveys

Robotic telescopes and digital sky surveys are complex research instruments, with sophisticated hardware and photometric pipelines that collect, analyze, store, and provide access to the data. Due to their complexity, it is difficult to fully verify that the data provided by these instruments is completely symmetric. On the other hand, it is also difficult to identify a possible flaw that would exhibit itself in the form of differences between the number of clockwise and counterclockwise galaxies. Moreover, the observation reported here is consistent across five major instruments that are independent of each other, while it is difficult to think of a flaw that would exhibit itself in this form in one instrument, it is definitely difficult to propose a flaw that is consistent across several different unrelated instruments.

### 4.4. Multiple Objects That Are Part of the Same Galaxy

Digital sky surveys use automatic object detection, and can identify multiple photometric objects in the same galaxy as independent galaxies. Such objects can include satellite galaxies, merging systems, large star clusters, large detached segments, and more. For all datasets used in this study, objects that are part of the same system were removed. That was done by identifying and removing objects that had another object in the database at a distance of 0.01° or less. That was not done for the galaxies imaged by Hubble Space Telescope, where the field is much smaller, making the angular distance between the objects far shorter compared to Earth-based surveys, which have far larger footprints.

Even if duplicate photometric objects are present, they are expected to be distributed evenly between clockwise and counterclockwise spiral galaxies. That even distribution is not expected to exhibit an asymmetry. Empirical experiments of adding artificial duplicate objects showed that adding duplicate objects does not result in asymmetry in the distribution of galaxy spin directions [165].

These experiments used $\sim 7.7 \times 10^4$ spiral galaxies, and assigned each galaxy with a random spin direction. By duplicating these galaxies, the effect of adding duplicate objects was observed. The results showed that adding duplicate objects did not turn a randomly distributed dataset into statistically significant asymmetry, unless a very large amount of $\sim 500\%$ of duplicated objects are added [165]. That experiment showed that even if duplicated objects existed in the dataset, it could not have led to the asymmetry.

### 4.5. Atmospheric Effect

It is difficult to think of an atmospheric effect that can flip the spin direction of a galaxy as seen from Earth. Additionally, the difference between the number of galaxies spinning clockwise and galaxies spinning counterclockwise is always determined by using galaxies observed in the same image of the same field, an atmospheric effect that impacts clockwise spiral galaxies will have the same impact on counterclockwise galaxies. Therefore, if such an atmospheric effect existed, it is not expected to lead to the asymmetry between the number of galaxies with opposite spin directions. Furthermore, one of the datasets used in this study is a dataset of spiral galaxies collected by HST. As a space-based instrument, HST is not affected by the atmosphere.

### 4.6. Spiral Galaxies with Leading Arms

While most spiral arms are trailing, the arms of a spiral galaxy can in some cases be leading. A known case of a spiral galaxy with a leading arm is NGC 4622 [202]. If galaxies

with trailing arms are not evenly distributed between clockwise and counterclockwise galaxies, that can result in asymmetry in spin directions of spiral galaxies in the observed Universe. For instance, if a high percentage of clockwise galaxies have leading arms, a, large-scale analysis of the distribution of galaxy spin directions would show an asymmetry, with a higher number of counterclockwise galaxies.

However, spiral galaxies with leading arms are a minority among spiral galaxies. Furthermore, spiral galaxies with leading arms are expected to be equally prevalent among clockwise and counterclockwise galaxies. For instance, the asymmetry between clockwise and counterclockwise galaxies can be defined by Equation (4)

$$A = \frac{(L_{cw} + T_{ccw}) - (L_{ccw} + T_{cw})}{L_{cw} + T_{ccw} + L_{ccw} + T_{cw}}, \tag{4}$$

where $T_{cw}$ and $T_{ccw}$ are the number of clockwise and counterclockwise galaxies with trailing arms, and $L_{cw}$ and $L_{ccw}$ are the number of clockwise and counterclockwise galaxies with leading arms. If $L_{cw} \simeq L_{ccw}$, the asymmetry $A$ can be defined by Equation (5). Because $L_{cw} \geq 0$ and $L_{ccw} \geq 0$, higher presence of spiral galaxies with leading arm can make the asymmetry $A$ smaller, but not higher.

$$A = \frac{T_{cw} - T_{ccw}}{L_{cw} + T_{cw} + L_{ccw} + T_{ccw}} \tag{5}$$

### 4.7. High Asymmetry in a Specific Part of the Sky

As was shown in an experiment with artificial data [165], high asymmetry in a specific small part of the sky can lead to a statistically significant dipole axis that peaks exactly at the center of the region where the distribution is asymmetric. Even if the distribution of spin directions in the rest of the sky is random when fitting the distribution into a dipole axis the presence of asymmetry in a small part of the sky can exhibit itself in the form of a statistically significant dipole axis. That is, even a relatively small region in the sky in which the number of clockwise and counterclockwise galaxies is significantly different can lead to asymmetry when analyzing the entire sky.

The results of this study show that the asymmetry is inverse in the opposite hemisphere, and therefore the axis cannot be driven by the asymmetry in a single part of the sky. Figure 5 shows consistency in the asymmetry in different 30° RA slices, showing that neighboring slices normally have a closer magnitude of asymmetry, and the asymmetry in each RA slice is inverse to the corresponding RA slice in the opposite hemisphere. Additionally, Table 8 shows that in the part of the sky centered at the COSMOS field the distribution is asymmetric. Obviously, the COSMOS field was not selected for its distribution of galaxy spin, and therefore can be considered an arbitrary part of the sky. The observation that the asymmetry exists in a part of the sky that was chosen arbitrarily, is another indication that the asymmetry exists in just one certain part of the sky.

**Table 8.** The distribution of clockwise and counterclockwise galaxies in the HST COSMOS field, and in the larger field around COSMOS in SDSS, Pan-STARRS, and DESI Legacy Survey. The P values are the one-tail accumulated binomial distribution probabilities when the probability of a galaxy spinning in a certain direction is 0.5.

| Sky Survey | # Clockwise Galaxies | # Counterclockwise Galaxies | $p$ Value |
|---|---|---|---|
| COSMOS (HST) | 3116 | 2965 | 0.027 |
| Pan-STARRS | 183 | 150 | 0.04 |
| SDSS | 349 | 308 | 0.06 |
| DESI | 2540 | 2410 | 0.03 |

As shown in Table 7, analysis of different sky surveys shows dipole axes that peak well within one sigma error from each other. That is true also in the case of sky surveys that their footprints do not overlap, such as DES and SDSS. The consistency between non-overlapping sky surveys provides another indication that the profile is not driven by the strong alignment of spin directions in a certain specific part of the sky, but could be related to the entire sky, while the profile shown here can be driven by alignment in galaxy spin directions in certain specific small parts of the sky, the observations shown in this paper suggest that it is possible that the asymmetry profile is related to the entire sky.

### 4.8. Dependence between Morphological Analysis and the Redshift

The galaxies used in this experiment have different redshifts. Because galaxies with higher redshifts tend to be dimmer and smaller, the morphological analysis of the galaxies might become more difficult for galaxies with high redshift. The galaxies used in this experiment are limited by radius and magnitude so that all galaxies are relatively large and bright. Previous analysis of the dependence between the morphological analysis and redshift has shown mild dependence on the redshift when the magnitude and radius of the galaxies are controlled [162].

Additionally, previous experiments showed that the magnitude of the asymmetry increases when the redshift gets higher [162,167]. For instance, Table 9 shows the magnitude of the asymmetry in different redshift ranges in the RA range of (120°, 210°). The analysis is based on SDSS galaxies with spectra, and reported in [162]. The table shows higher asymmetry in higher redshift ranges, and similar observations were made for other parts of the sky, while in the RA range of (120°, 210°) the number of galaxies spinning counterclockwise grows with the redshift, in other parts of the sky the number of galaxies spinning clockwise grows, indicating that the results are not driven by systemic inaccuracies of the annotation.

**Table 9.** The number of clockwise and counterclockwise galaxies in SDSS, separated into different redshift ranges. All galaxies are within the RA range of (120°, 210°). The results are taken from [162].

| z | cw | ccw | $\frac{cw - ccw}{cw + ccw}$ | p Value |
|---|---|---|---|---|
| 0–0.05 | 3216 | 3180 | 0.0056 | 0.698 |
| 0.05–0.1 | 6240 | 6270 | −0.0024 | 0.4 |
| 0.1–0.15 | 4236 | 4273 | −0.0043 | 0.285 |
| 0.15–0.2 | 1586 | 1716 | −0.039 | 0.008 |
| 0.2–0.5 | 2598 | 2952 | −0.064 | $1.07 \times 10^{-6}$ |
| Total | 17,876 | 18,391 | 0.493 | 0.0034 |

As explained in Section 4.1, a higher error in the annotations of the galaxies is expected to lead to a lower magnitude of the asymmetry. The increase in the asymmetry as the redshift gets higher suggests that the change is not driven by the error of the annotation of galaxies with higher redshift. Furthermore, the effect of the redshift is expected to have a similar impact in all parts of the sky, and have a similar effect on galaxies that spin clockwise and galaxies that spin counterclockwise. As also mentioned above, while the increase in the magnitude was observed in other parts of the sky, the direction of the asymmetry flips, and in other parts of the sky the number of clockwise galaxies increases.

Furthermore, because the selection of the galaxies is done by the apparent magnitude and radius, in higher redshift ranges the galaxies that are selected are larger than in lower magnitudes. That could indicate that the spin asymmetry correlates with the absolute magnitude and size of the galaxies, and consistently their stellar mass. In that case, larger galaxies that could have been the outcome of previous mergers could show higher asymmetry in the distribution of their spin directions. However, an experiment of using galaxies of different radii showed no significant change in the magnitude of the asymmetry when changing the size of the galaxies. That experiment is described in Section 3 in [162], and might indicate that the asymmetry changes with the redshift of the galaxies rather

than with a higher stellar mass. That observation is aligned with the contention that galaxy mergers, which also increase the stellar mass of the galaxy, are not expected to lead to the less random distribution of the spin directions [147]. The correlation between the large-scale asymmetry in spin directions and time provides certain evidence that the spin direction alignment is driven by the cosmic initial conditions, and becomes more stochastic in time, possibly through galaxy mergers. That contention, however, is based on galaxies at relatively low redshift, and future studies with galaxies at higher redshifts will be required to better profile the correlation.

### 4.9. Bias in the Selection of the Galaxies

The analysis does not rely on any previously used catalog, and therefore unknown biases from previous catalogs designed for other purposes cannot be carried over to the analysis shown here. As discussed in Section 2.1, most of the galaxies do not have an identifiable spin direction, and therefore are rejected from the analysis. That is a limitation of any algorithm since the absence of an identifiable spin direction does not mean that the galaxy does not spin. For instance, Figure 10 shows examples of three galaxies imaged by both SDSS and HST, while the images taken by SDSS show no identifiable spin directions, the HST images show very clear spin patterns and identifiable counterclockwise spin directions.

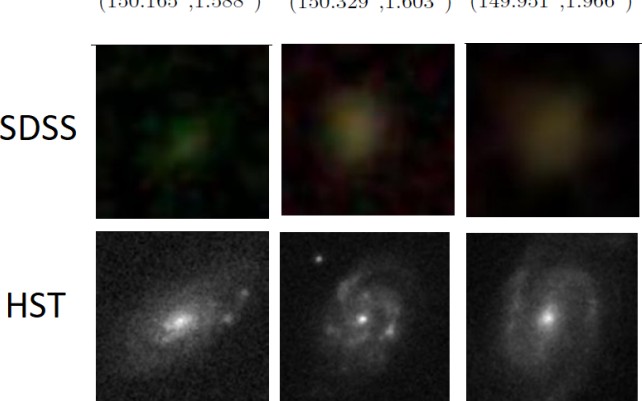

**Figure 10.** Examples of galaxies imaged by both HST and SDSS, while in images acquired by SDSS the galaxies do not seem to have an identifiable spin direction, the HST images of the exact same galaxies show that these galaxies have counterclockwise spin patterns. The equatorial celestial coordinates of each galaxy are specified at the top of each column.

A bias in the selection of galaxies such that, for instance, a higher number of clockwise galaxies are rejected from the analysis, can lead to asymmetry in the algorithm. Since the algorithm is symmetric, and is not based on machine learning or human analysis, it is expected that the number of galaxies that spin clockwise but are rejected from the analysis is the same, within statistical error, as the number of counterclockwise galaxies rejected from the analysis.

However, assuming that the algorithm does have a certain unknown bias, and it tends to reject more galaxies that spin towards a certain direction, the biased asymmetry $A$ in a certain field is determined by Equation (6)

$$A = \frac{R_{cw} \cdot d_{cw} - R_{ccw} \cdot d_{ccw}}{R_{cw} \cdot d_{cw} + R_{ccw} \cdot d_{ccw}}, \tag{6}$$

where $R_{ccw}$ and $R_{cw}$ are the number of galaxies that indeed spin counterclockwise and clockwise, respectively. $d_{ccw}$ is the fraction of counterclockwise galaxies that the algorithm detects their spin direction, and therefore are used in the analysis. Similarly, $d_{cw}$ is the fraction of clockwise galaxies whose spin direction is detected by the algorithm, and these galaxies make the set of clockwise galaxies in the analysis.

Assuming that the real distribution of the spin directions in the dataset is fully random, $R_{cw}$ is expected to be equal (within statistical error) to $R_{ccw}$. In that case, the observed asymmetry $A$ can be expressed as shown by Equation (7)

$$A = \frac{(d_{cw} - d_{ccw})}{(d_{cw} + d_{ccw})}. \tag{7}$$

The algorithm is static, is not trained by galaxies taken from different parts of the sky, and therefore does not change during the analysis. If the algorithm is biased, $d_{cw}$ is expected to be different from $d_{ccw}$. Because the annotation algorithm does not change during the analysis, $d_{cw}$ and $d_{ccw}$, even if they are biased, are expected to be the same. That is, if the distribution of the spin direction is fully random, but $d_{cw}$ or $d_{ccw}$ are biased, the asymmetry $A$ will have a non-zero value that is consistent in all parts of the sky.

However, as shown in Table 3, Figure 5, and other previous work [161,162,166,167], the asymmetry $A$ is different in different parts of the sky, and its sign flips in opposite hemisphere. The differences between the asymmetry in different parts of the sky are statistically significant, which means that the distributions $R_{cw}$ and $R_{ccw}$ are not fully random.

## 5. Previous Studies That Show Different Conclusions

While several previous studies mentioned in Section 1 provided results suggesting that the large-scale distribution of galaxy spin directions is not necessarily random, other studies used similar approaches to reach opposite conclusions. Here, I analyze these studies to identify reasons for the differences.

One of the first attempts to study the distribution of spin directions of spiral galaxies was based on manual selection of the galaxies and manual annotation of their spin directions [203]. The analysis made use of a dataset of $\sim 6.5 \times 10^3$ galaxies, showing that the distribution of their spin directions was random. However, the small size of that dataset did not allow to identify the asymmetry with statistical significance. For instance, assuming that the magnitude of the asymmetry is 1% as shown in this paper, $2.7 \times 10^4$ galaxies are required to show a one-tailed P value of $\sim 0.05$. Even when assuming a larger magnitude of 2%, at least $7 \times 10^3$ galaxies are required to provide one-tailed probability of $p \simeq 0.048$. That shows that a dataset of merely $\sim 6.5 \times 10^3$ objects is not sufficiently large to show a statistically significant signal of asymmetry in the distribution of galaxy spin directions.

Another attempt to study the distribution of spin directions of spiral galaxies used manual annotations of galaxies by using the Galaxy Zoo platform, providing access to crowdsourcing done by non-scientist volunteers [204]. By using the practice of crowdsourcing, the experiment could use a large number of annotated galaxies, as the large number of volunteers increased the overall throughput of the annotation. The primary weakness of that approach was that the annotations were heavily impacted by human bias [204]. The use of manual annotation by anonymous volunteers led to inaccuracy in the annotations. More importantly, the bias was systematic. Because the bias was not known when the project was designed, the galaxies were not mirrored randomly to balance the human bias.

When the bias was noticed, a small number of galaxies were re-annotated such that the galaxy images were also mirrored. That experiment showed that 5.525% of the galaxies were annotated as clockwise, and 6.032% of the galaxies were annotated as counterclockwise. When the galaxy images were mirrored, the numbers changed to 5.942% clockwise, and 5.646% counterclockwise. These results are shown in Table 2 in [204]. That is, after the galaxies were mirrored, the frequency of galaxies spinning counterclockwise dropped by $\sim 1.5\%$, and the frequency of galaxies spinning clockwise increased by $\sim 2\%$. The asymmetry of 1–2% agrees in both its direction and its magnitude with the asymmetry described in [162]. The analysis of [162] used SDSS galaxies with spectra, and therefore the experiment described in [162] used the same footprint and distribution of the Galaxy Zoo galaxies analyzed in [204], making it relevant for comparison to Galaxy Zoo.

Because just a small number of galaxies were mirrored, the size of the dataset of annotated galaxies was relatively small, at $\sim 1.1 \times 10^4$ galaxies [204]. Due to the small number of annotated galaxies, the results were not statistically significant. At the same time, the asymmetry shown in [204] also does not conflict with the asymmetry shown in this study or in [162]. It was also argued that the distribution of Galaxy Zoo annotations of the directions toward which galaxies spin is not necessarily random [205].

An analysis based on automatic annotation [206] using the *SPARCFIRE* algorithm [207,208] showed that the spin directions of galaxies annotated by Galaxy Zoo are distributed randomly. When just applying the automatic annotation to galaxies annotated as spiral by Galaxy Zoo, the asymmetry was statistically significant, with $2.52\sigma$ or stronger. These results are summarized in Table 2 of [206]. A possible reason that can explain the statistically significant asymmetry is that the annotation of spiral galaxies by Galaxy Zoo volunteers was the reason of the asymmetry. That is a new kind of bias that was not reported in [204].

To correct for that bias, the selection of spiral galaxies was performed by using a machine learning algorithm. The machine learning algorithm was trained with two classes of galaxies—elliptical and spiral. Its goal was to select spiral galaxies automatically, and reject elliptical galaxies from the analysis. To ensure that the algorithm does not select a higher number of clockwise galaxies or counterclockwise galaxies as spiral, the training set of spiral galaxies was made of 50% spiral galaxies that spin clockwise, and 50% spiral galaxies that spin counterclockwise. That is, the machine learning algorithm that selected the spiral galaxies was trained with one class of elliptical galaxies, and another class of spiral galaxies. The class of spiral galaxies contained an equal number of clockwise and counterclockwise galaxies to ensure that a higher population of a certain spin direction in the training set does not lead to a selection of more spiral galaxies that spin in that direction, while machine learning is often difficult to fully analyze [169,170], given the known limitations of machine learning the design of the experiment seems sound and unbiased.

However, in addition to the balanced training set, the machine learning algorithm was also designed such that all attributes that identified asymmetry in the galaxy spin directions were specifically removed from the data. As stated in [206] "We choose our attributes to include some photometric attributes that were disjoint with those that Shamir (2016) found to be correlated with chirality, in addition to several SPARCFIRE outputs with all chirality information removed".

The removal of attributes that identify between clockwise and counterclockwise galaxies naturally led to a dataset that is more symmetric in spin directions. That is, when removing the attributes that can identify the spin direction of the galaxy, the machine learning algorithm produced a dataset that is more symmetric when applying an algorithm to annotate the galaxies by their spin direction. Although it is difficult to predict subtle biases in machine learning systems, theoretically that machine learning system was expected to be symmetric between clockwise and counterclockwise galaxies. Still, for showing randomness in the spin directions of the galaxies, the removal of attributes that identify the spin direction was required.

When not correcting for the attributes that correlate with galaxy spin direction asymmetry, the asymmetry between the number of clockwise and counterclockwise galaxies is with statistical significance of $2.52\sigma$ or stronger when using Galaxy Zoo galaxies classified as a spiral. These results are specified in Table 2 in [206]. The statistical significance changes with the threshold of the selection of spiral galaxies, but in all cases it was statistically significant. These results are also in agreement with previous analysis of SDSS galaxies as reported in [162].

Another study that showed opposite results used the dataset of [209] and argued that the asymmetry is the result of "duplicate objects" [210]. When removing the "duplicate objects" to create a "clean" dataset, the signal drops to $0.29\sigma$. As the abstract claims "The actual dipole asymmetry observed for the "cleaned" catalog is quite modest, $\sigma_D = 0.29$".

However, the dataset that was used in [209] was used for studying differences in the photometry objects that spin in opposite directions. The [209] paper does not make any claim for the existence of a dipole axis or any other axis formed by the spin directions of spiral galaxies. No claim for an axis is made regarding that dataset in any other paper. When using the same dataset to analyze the existence of a dipole axis in galaxy spin directions, photometric objects that are part of the same galaxy such as satellite galaxies or galaxy mergers become "duplicate objects". However, [209] does not make any attempt to study any kind of axis formed by the asymmetry in the distribution of galaxy spin directions. As mentioned above, no such claim was made in any other paper.

The more interesting question in the sense of the distribution of galaxy spin directions in the local Universe is why a "clean" dataset exhibited random distribution. That can be explained by careful analysis of the statistical analysis of the exact same "clean" dataset used in [210].

The statistical significance of $0.29\sigma$ reported in the abstract of [210] was the result of an experiment such that the redshift of all galaxies was limited to z < 0.1. As explained in [162], limiting the redshift leads to a weaker statistical signal of the dipole. For instance, when using just galaxies with the redshift of z < 0.15, the statistical signal of the dipole axis becomes insignificant. That can also be seen in Tables 3 and 5–7 in [162]. The tables show that the statistical significance of the distribution becomes insignificant when the redshift ranges are lower. Therefore, the low statistical significance when limiting the redshift to z < 0.1 reported in [210] is in full alignment with previous studies [162].

More importantly, unlike the analysis used in this paper, the analysis of [210] applied a three-dimensional analysis, where the position of each galaxy is determined by its galactic coordinates and its redshift. Because the galaxies used in [209] do not have spectra, the analysis shown in [210] used the "measured redshift" of each galaxy, which is in fact the photometric redshift of the galaxies taken from the photometric redshift catalog of [211], while spherical coordinates of galaxies are considered accurate, the photometric redshift is highly inaccurate, and can also be systematically biased. The inaccuracy of the photometric redshift of the [211] catalog is ~18.5% [211]. That error is substantially higher than the magnitude of the asymmetry, which is ~1%. Therefore, the error of the photometric redshift is expected to add inaccuracy to the analysis, and consequently weaken the observed signal.

When not limiting the galaxies to the photometric redshift of less than 0.1, and when applying a statistical analysis that does not rely on the photometric redshift, it is clear that the distribution of the galaxy spin directions in that dataset is not random. For instance, a very simple binomial distribution analysis of the data shows that the galaxies can be separated into two hemispheres, such that one has a higher number of galaxies spinning clockwise, while the opposite hemisphere has a higher number of galaxies spinning counterclockwise. The exact same "clean" dataset of 72,888 galaxies used in [210] can be accessed at https://people.cs.ksu.edu/~lshamir/data/assym_72k (accessed on 2 June 2022).

Table 10 shows the distribution of galaxies spinning clockwise and galaxies spinning counterclockwise in the hemisphere centered at $\alpha = 340°$, and in the opposite hemisphere centered at $\alpha = 160°$. As the table shows, the asymmetry in the distribution of the spin directions of spiral galaxies in the hemisphere centered at $\alpha = 160°$ is statistically significant. In the opposite hemisphere the asymmetry is not statistically significant. However, that hemisphere also shows a higher number of galaxies spinning counterclockwise, and therefore it is also not in disagreement with the asymmetry in the hemisphere centered at (RA = 160°) for the contention that the two hemispheres form a dipole axis. Since there are two hemispheres, the overall statistical significance of the distribution can be determined by applying a Bonferroni correction to the two-tailed P value of the distribution in the hemisphere centered at (RA = 160°). This provides a probability of ~0.01 of such distribution or stronger to happen by chance.

Monte Carlo simulation showed that such asymmetry or stronger was observed in just 70 runs out of a total of 100,000 attempts. That is a probability of ∼0.007. Code and data to reproduce the Monte Carlo simulation can be found at https://people.cs.ksu.edu/~lshamir/data/iye_et_al (accessed on 2 June 2022).

**Table 10.** The number of clockwise and counterclockwise galaxies in the exact same "clean" dataset of 72,888 galaxies used in [210]. The *p* values are the binomial distribution such that the probability of a galaxy to in a certain direction is random 0.5. The data are available at https://people.cs.ksu.edu/~lshamir/data/assym_72k/ (accessed on 2 June 2022).

| Hemisphere (RA) | # Clockwise | # Counterclockwise | $\frac{\#Z}{\#S}$ | *p* (One-Tailed) | *p* (Two-Tailed) |
|---|---|---|---|---|---|
| >250° ∪ <70° | 13,660 | 13,749 | 0.9935 | 0.29 | 0.58 |
| 70°–250° | 23,037 | 22,442 | 1.0265 | 0.0026 | 0.0052 |

Reproduction of the analysis described in [210] without using the photometric redshift shows a statistically significant dipole axis. Figure 11 shows the statistical significance of fitting the galaxy spin direction into a form of a dipole axis from each possible $(\alpha, \delta)$ combination. The dataset is the exact same "clean" dataset of 72,888 galaxies used in [210], but without using the photometric redshift to determine the position of each galaxy. The most likely axis peaks at $(\alpha = 165°, \delta = 40°)$, with statistical significance of $2.14\sigma$. That statistical signal is not normally considered random. Because the galaxies used in [209] are relatively bright (i magnitude < 18) and large (Petrosian radius < 5.5′) galaxies, these galaxies also have lower redshifts. As shown in [162], a weaker signal in the asymmetry is expected when the redshift is lower. However, in any case, the statistical signal of the dipole axis is $>2\sigma$, and therefore is not considered random. Code and data to reproduce the analysis are available at https://people.cs.ksu.edu/~lshamir/data/iye_et_al (accessed on 2 June 2022).

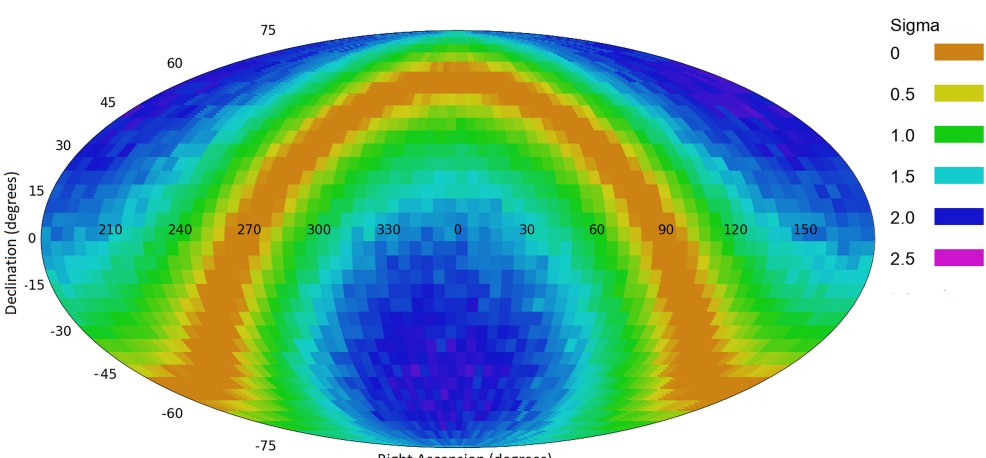

**Figure 11.** The $\chi^2$ statistical significance of a dipole axis in the spin directions of the galaxies from different $(\alpha, \delta)$ combinations in the exact same dataset used by [210]. Code and data to reproduce the analysis are available at https://people.cs.ksu.edu/~lshamir/data/iye_et_al (accessed on 2 June 2022).

Figure 12 displays the statistical significance of the dipole axis from different $(\alpha, \delta)$ combinations in the sky after assigning the galaxies random spin directions. The analysis shows a much lower statistical significance of less than $1\sigma$.

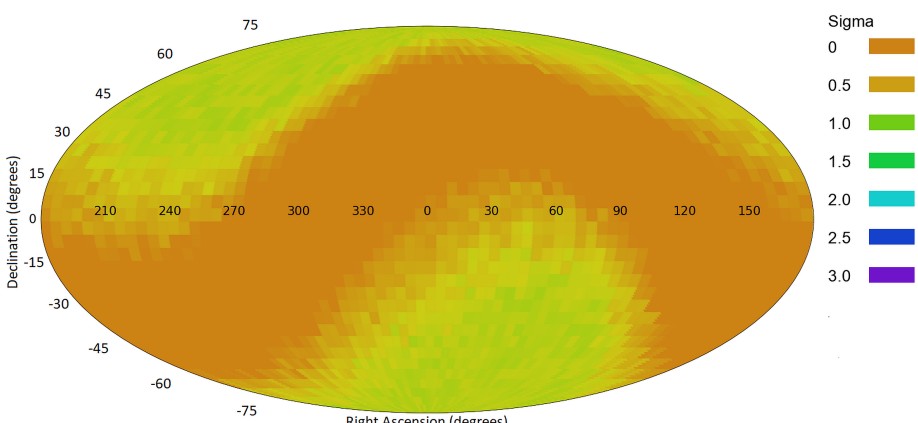

**Figure 12.** The $\chi^2$ statistical significance of a possible dipole axis from all possible integer $(\alpha, \delta)$ combinations when the spin directions of the galaxies are assigned randomly.

## 6. Discussion

The deployment of powerful digital sky surveys has revolutionized cosmology research by enabling the generation of very large astronomical databases, and reconstructing the local Universe in high detail. These databases allow addressing research questions that were not addressable in the recent past. One of the many examples of data-driven research questions that challenge the current cosmological models is the Hubble-scale structures [1–5] that were impractical to identify without the large datasets collected by autonomous digital sky surveys.

This paper examines the probe of the distribution of the spin directions of spiral galaxies. The study analyzes a large number of spiral galaxies to examine the nature of the distribution of spin directions in the context of the large-scale structure. The analysis is limited to the direction of the spin of each galaxy (clockwise or counterclockwise), and not the magnitude of the spin. Five different digital sky surveys covering different parts of the sky were analyzed, all showing patterns of asymmetry between the number of galaxies spinning in opposite directions. These results are aligned with multiple previous experiments, showing that the distribution of spin directions of spiral galaxies as observed from Earth might not be random [135,136,158–162,165–167,212]. Some previous studies showing opposite conclusions are discussed in Section 5, but these studies might not necessarily provide a definite proof of fully symmetric ratio.

While galaxies do not form from the same materials, the distribution of galaxies in the Universe is not random [55], and the Universe seems to be organized in walls and filaments that make the cosmic web. This large-scale structure is presumably linked to the underlying distribution of dark matter throughout the Cosmos. Yet, it is still unclear how matter moves in these cosmic filaments. A preferential spin axis can be induced if there is a tendency to move to the highest densities following a corkscrew motion.

Studies with smaller datasets of galaxies showed non-random spin directions of galaxies in filaments of the cosmic web [121–124,126,127,129–133]. Other studies showed alignment in the spin directions even when the galaxies are too far from each other to interact gravitationally [134–136], unless assuming modified gravity models [213,214] that explain longer gravitational span [215–217]. Modified gravity models were also proposed as an explanation to the $H_o$ tension [218].

One of the mature theories that explain the initiation of galaxy spin is the *tidal torque theory* [110–114,116–119,147]. According to tidal torque theory, the angular momenta of galaxies are directly linked to the initial conditions [147]. Another theory of galaxy spin is the merging of galaxies [144,145], or interactions between dark matter halos that do not merge but pass in close proximity to each other [146].

If galaxy mergers were the sole agent that initiates galaxy spin, the distribution of spin directions of galaxies would have been expected to be stochastic [147]. This study shows with a very large number of galaxies and several different sky surveys that the spin directions of galaxies is not stochastic, and therefore agrees with the contention that initial galaxy spins are more likely related to the cosmic initial conditions rather than the sole outcome of galaxy mergers.

The alignment of spin directions within filaments of the cosmic web has been observed to correlate with the mass [131], where galaxies at a lower mass were correlated with alignment with the parent structure, and higher mass correlated with perpendicular alignment to the structure. That can be explained by *flip-spin*, where initial galaxy angular momentum is directly related to the initial conditions, and galaxy mergers flip the spin [131]. The link between mass and alignment of spin directions was also observed in computer simulations [127,129,137,139,140,150–156,219]. The strength of that correlation has been linked to the galaxy stellar mass and the color of the galaxies [155,220]. It has been proposed that that link is associated with halo formation [154], which can lead to a link between the large-scale structure of the Early Universe and the spin directions [142].

The analysis described in this paper shows cosmological-scale anisotropy that spans over a portion of the sky that is far larger than any known supercluster, filament, or wall in the large-scale structure. That suggests that the link between the cosmic initial conditions and the alignment of galaxy spin directions might be exhibited by the entire local Universe, and is not limited to specific walls or filaments.

In addition to alignment in spin directions of galaxies, observations of large-scale alignment in spin directions were observed with quasars [221]. Position angle of radio galaxies also showed large-scale consistency of angular momentum [222]. These observations agree with observations made with datasets such as the Faint Images of the Radio Sky at Twenty-centimetres (FIRST) and the TIFR GMRT Sky Survey (TGSS), showing a large-scale alignment of radio galaxies [223,224]. In general, the distribution of galaxies in the Universe is far from random [55,225,226].

The contention of Hubble-scale anisotropy [227] has been proposed based on observations of the CMB distribution [12–16,19]. The cosmological-scale axis exhibited by the CMB agrees with other large-scale asymmetry axes formed by probes such as dark energy and dark flow [13]. Other CMB anomalies include the quadrupole-octopole axis alignment [228–232], point-parity asymmetry [233,234], CMB asymmetry between opposite hemispheres [7,14,235], and the cold spot in the cosmic microwave background [191–195], while it has been suggested that these observations are not necessarily of statistical significance [236], they can also be considered as observations that conflict with ΛCDM cosmology [57].

The provocative contention of the existence of a Hubble-scale axis in the Universe is aligned with several cosmological models as described in Section 1. These theories shift from the standard model. One of the notable alternative theories is black hole cosmology [87–90,98,100,237]. One of the initial observations that agree with the contention of a black hole universe is the agreement between the Hubble radius and the Schwarzschild radius of a black hole when the mass is the mass of the Universe [100]. Another supporting observation is the accelerated inflation of the Universe, which according to black hole cosmology does not require the assumption of dark energy. It should be mentioned that the first models of black hole cosmology were proposed before the inflated acceleration of the Universe was discovered. Black hole cosmology can be classified under the category of multiverse [238–241], which is one of the older cosmological theories [242,243].

Because stellar black holes are born spinning [91–96], a universe in the interior of a stellar black hole is expected to have a major axis inherited from its host black hole. While stellar black holes are formed from the gravitational collapse of stars, the formation of supermassive black holes is still not fully understood [244]. A possible explanation is that supermassive black holes are formed from the collapse of supermassive stars in the early Universe [244,245]. Another explanation is that the merging of smaller black holes can lead

to larger black holes [246], and in that case the gravitational interaction of the merging of black holes can lead to spin [246]. It has also been proposed that supermassive black holes are the result of *direct collapse* of gas clouds into a supermassive black hole without an intermediate stage of a star [247,248]. According to that theory, supermassive black holes are also expected to spin [247]. Regardless of the theory, empirical observations of supermassive black holes showed that these black holes spin [249], as initially observed with the supermassive black hole of NGC 1365 [250].

A possible universal pattern of galaxy spin directions can also be related to the proposed existence of a Universal force field [251]. The observation that galaxies in opposite lines of sight show opposite spin directions also agrees with cosmology driven by longitudinal gravitational waves [252], according to which each galaxy at a certain distance from Earth is expected to have an antipode galaxy under the same physical conditions, but accelerating oppositely [252]. The link between that theory and the observations discussed here might be challenged by the fact that the magnitude of the observed asymmetry is mild, and does not support the full separation of the galaxy population into pairs of antipode galaxies.

Another direction of explanation that should be considered is that the observation is related to the internal structure of galaxies. The rotation direction of extra-galactic objects can affect photons [253], making objects rotating in opposite directions visually different when observed from Earth. For instance, due to special relativity, galaxies with a spin direction orthogonal to the spin direction of the Milky Way are supposed to be visually different to an Earth-based observer from identical galaxies that spin in the opposite direction. A galaxy with a rotational velocity of $v_r$ relative to a Milky Way observer will have a Doppler shift of its bolometric flux of $F_o = (1 + 4 \cdot \frac{V_r}{c})$, where $F_0$ is the flux of the galaxy when it is stationary compared to a Milky Way observer, and $c$ is the speed of light [254]. Assuming rotational velocity of $2 \times 220$ km $\cdot$ s$^{-1}$ of the observed galaxy relative to the Milky Way, $\frac{v}{c}$ is ~0.0015. That means that $\frac{F}{F_0}$ is $\simeq$1.0059, and will lead to a maximum magnitude difference of $-2.5 \log_{10} 1.0059 \simeq 0.0064$. That subtle magnitude difference is not expected to affect the observation, and is far smaller than the observed differences [164].

However, the physics of galaxy rotation are still not fully understood. As it is clear that galaxy rotation does not follow Newtonian physics (when assuming that most matter of the galaxy is not dark), the leading assumption to explain the puzzling conflict is that the mass of galaxies is dominated by dark matter [255–257]. The assumption of dark matter is a key working assumption in most current standard cosmological models, but the nature of dark matter has not been fully proven and profiled. Other leading explanations include the contention that galaxy rotation does not follow the known physics, but driven by modified physics that applies to galaxy rotation [258–264]. These theories also have relativistic expansions [265–271].

Driven by robotic telescopes and autonomous data acquisition instruments, the information revolution has enabled a new way to observe the Universe. Clearly, further research will be needed to verify and profile the observation. However, the observation by five different sky surveys, all showing a statistically significant dipole axis within $1\sigma$ error from each other, should lead to the consideration of the question of whether the spin directions of spiral galaxies as seen from Earth is indeed fully random.

**Funding:** This study was supported in part by USA National Science Foundation grants AST-1903823 and IIS-1546079.

**Data Availability Statement:** Not applicable.

**Acknowledgments:** I would like to thank the two knowledgeable reviewers for excellent comments that greatly helped to improve the paper. I would like to thank Ethan Nguyen for retrieving and organizing the image data from the DESI Legacy Survey.

**Conflicts of Interest:** The author declare no conflict of interest.

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
