# Peer review of "Asymmetry in Galaxy Spin Directions—Analysis of Data from DES and Comparison to Four Other Sky Surveys"

_universe, doi:10.3390/universe8080397_

Round 1
Reviewer 1 Report
This manuscript presents an analysis of galaxy spin orientations on the sky. It finds a small but significant systematic orientation, which is surprising and very important if correct. The analysis is solid so should be published; I have only a few minor comments for the author's consideration.
1. Annotation of the spin direction cannot be assigned for the majority of galaxies. This is unsurprising given that the method relies on the automated detection of spiral features, while many galaxies are irregular or elliptical (and without a definitive spin axis). It might help to address the expectation for how often this should work early on given that it often does not. I can't imagine any systematic that would lead to a false signal for lacking the non-annotated galaxies, but it is a foible that should be addressed.
2. Is it possible to assign a spin vector as projected on the sky? This paper relies on determinations of clockwise and counterclockwise spin directions, which is fine, so what I suggest is beyond the scope of this paper. But there is additional information in the images, which have some orientation (position angle) of the major axis of each galaxy on the sky. Combined with the CW or CCW annotation, this defines a vector in the 2 dimensions of the sky. One might even be so bold as to use the minor-to-major axis ratio to assign an inclination and hence a 3D vector. If the detected effect is real and not some quixotic observational systematic, then I would expect the significance of the vector signal to be higher than that of the simple CW or CCW annotations.
3. At the beginning of section 3, it is noted that the standard error is the inverse square root of the total number of galaxies. Is that the right measure given that many galaxies remain unannotated? I would think the proper statistic would be sqrt(CW+CCW) rather than sqrt(N). How does that impact the p-values given in the tables?
Author Response
This manuscript presents an analysis of galaxy spin orientations on the sky. It finds a small but significant systematic orientation, which is surprising and very important if correct. The analysis is solid so should be published; I have only a few minor comments for the author's consideration.
--Author response: Thank you for the time you spent reading and commenting in the manuscript, and for the comments and the encouraging words.
- Annotation of the spin direction cannot be assigned for the majority of galaxies. This is unsurprising given that the method relies on the automated detection of spiral features, while many galaxies are irregular or elliptical (and without a definitive spin axis). It might help to address the expectation for how often this should work early on given that it often does not. I can't imagine any systematic that would lead to a false signal for lacking the non-annotated galaxies, but it is a foible that should be addressed.
--Author response: Thank you for the comment. First of all, a new subsection (Section 4.9) that is dedicated to the selection of the galaxies has been added. Figure 10 shows that no matter what, there will always be galaxies that have a spin direction but are not annotated. The solution is to address that symmetrically, so that the number of galaxies being unidentified is the same in both spin directions. That is now addressed and explained with simple mathematical analysis in the new Section 4.9. A note has been added also to Section 2 with a link to Section 4.9 that discusses the selection.
- Is it possible to assign a spin vector as projected on the sky? This paper relies on determinations of clockwise and counterclockwise spin directions, which is fine, so what I suggest is beyond the scope of this paper. But there is additional information in the images, which have some orientation (position angle) of the major axis of each galaxy on the sky. Combined with the CW or CCW annotation, this defines a vector in the 2 dimensions of the sky. One might even be so bold as to use the minor-to-major axis ratio to assign an inclination and hence a 3D vector. If the detected effect is real and not some quixotic observational systematic, then I would expect the significance of the vector signal to be higher than that of the simple CW or CCW annotations.
--Author response: That is definitely a good idea. I saw some previous work done by analyzing the position angle such as (Taylor et al., 2016) that is now mentioned in the Discussion section. It will be very interesting to combine alignment in position angle with alignment in spin direction. That, however, requires a completely new analysis, and is a topic in itself. That is definitely not practical in the 10 days I was given to revise and re-submit, but the paper is currently already nearly 20,000 word long (37 pages). Adding another analysis that is different from everything else in the paper will make it even longer and more complex. That will be something for future work, when the position angles of all these galaxies are computed and analyzed.
- At the beginning of section 3, it is noted that the standard error is the inverse square root of the total number of galaxies. Is that the right measure given that many galaxies remain unannotated? I would think the proper statistic would be sqrt(CW+CCW) rather than sqrt(N). How does that impact the p-values given in the tables?
--Author response: Thank you for the comment! You are obviously correct, and that is what is done in the analysis. The problem is that N was defined in the paper as “the total number of galaxies in the sky region”, which is misleading (and also impossible because we need an infinitely strong telescope to see *all* galaxies in a certain sky region). But N is the total number of galaxies in the sky region that were annotated. It is obviously N=cw+ccw, otherwise it would not have been the standard error. The has been corrected in the manuscript. Thank you for the comment.
Reviewer 2 Report
Report: Asymmetry in galaxy spin directions – analysis of data from DES and comparison to four other sky surveys
Author: Lior Shamir
I would like to thank you for the opportunity to revise this manuscript. The article is in good shape and a very interesting topic transversal to extragalactic astronomy and the Universe's large-scale structure. I understand the humungous work done here and want to congratulate you on this effort. Several possible biases were analysed in the text. First, I would like to recommend modifying the text and clarifying some parts, especially for people not from the area.
My comments are as follows:
I understand that the focus of the article is on the large-scale structure of the Universe, but I missed a more general introduction with citations and a connection with galaxy formation and evolution. For instance, how the spin of galaxies is an essential quantity to understanding the evolution of galaxies over cosmic time. Also, it would be beneficial for the article to know whether it is expected from galaxy formation and evolution scenarios that galaxies on large scales are organised within a network of filaments and walls and if it is expected that this large-scale structure produces (locally) anisotropy or not. For example, if this is true, I missed a discussion regarding how small portions of the sky could introduce biases in the analysis.
Indeed, when revising the tidal torque theory (Hoyle 1949; Peebles 1969; Doroshkevich 1970; White 1984; Catelan & Theuns 1996; Lee & Pen 2000, 2001, and Schäfer 2009, for a review), it has been proposed that galaxies acquire angular momentum via the tidal field from surrounding galaxies. In the Lambda Cold Dark Matter (ΛCDM) paradigm, galaxies are first formed in a dark matter halo and evolve through continuous merging and interaction with nearby galaxies while gaining or losing mass and angular momentum. I wonder if, after establishing and correcting for possible biases, we can find the winding sense of the galaxies to be consistent or not with statistical isotropy.
I've also seen that previous studies have shown that the spin of galaxies is correlated with environments, axes of large-scale structure, 2D correlation, and clustering (Porciani et al. 2002; Davis & Natarajan 2009; Tempel et al. 2013; Casuso & Beckman 2015; Codis et al. 2015; Lee et al. 2018b). Is there a possibility for these environmental biases in the current analysis? My point here is for the reader to understand a little bit better what is expected for galaxy spinning locally, connecting with galaxy formation/evolution and globally in terms of cosmological theories. This should be approached in the introduction.
Concerning the code Ganalyzer, I had difficulty understanding how the code works beneath the general mathematical explanation. I searched in the original Shamir (2011) paper and other articles, but still, there are some missing gaps. For instance, it seems that the images used as input for the code come from the automatic pipeline of the digital surveys, i.e. the coloured ones by combining different filters. For example, SDSS has created colour images by taking the middle three bands (g, r, i) and assigning those to the blue, green and red channels. And then, you have downloaded these coloured images as JPEG formats? Is this correct? It seems that this is the approach for others, such as in DES, for example. My question is: are you taking the artificially/pre-processed colour images from the digital surveys in JPEG format? I found this part somewhat awkward and confusing in the paper and propose that this part is clarified with more details. Additionally, If this is right, the approach presented in the paper could be inappropriate. It is well known that careful post-processing of imaging data is needed for higher accuracy with surface photometry techniques. For instance, Imaging data from surveys are usually provided in reduced form (e.g., 'u', 'g', 'r', 'i' and 'z' SDSS) after several pre-reduction steps:
1. Removal of instrumental/detector effects (bias, dark and flat-field, saturation, bad pixels);
2. Correction for cosmic ray hits;
3. Computation of calibration constants;
4. Approximate subtraction of the sky background through a low-degree polynomial;
5. Co-registration of images in some or all bands observed
However, the quality of reduction steps [4] & [5] is generally not sufficient for meticulous photometry. This is especially true for surface photometry profile (SBP) analysis and the determination of colour maps. For example, a minor error of typically less than 20% of the sky photon noise in the removal of the local sky background can result in a down-bending or flattening of SBPs in their outer parts, leading then to significant errors in the determination of e.g., the structural parameters of the disk component. Likewise, minor errors in the co-registration of images in two different bands can propagate into strong artefacts in colour maps computed by the division of these images. If JPEG images are used, then these are coloured images. As far as I understand, Ganalyzer provides radial intensity plots, nearly similar technique as SBPs, though not azimuthally averaged. If you are doing your own background removal and co-registration of images or even using one band at each time please clarify. Otherwise, I seriously wonder if your analyses affect the final conclusions, for instance, with the introduction of peaks/oscillations due to misalignments.
Connected with the previous paragraph, I also profoundly missed a detailed description of the datasets and what is used as input. The photometric bands or the combination of them used for each instrument. For instance, SDSS has five bands, is this analysis used in each one of the bands or in the colour image? I found the description of the datasets extremely poor and missing details. I understand that these may have been discussed in previous papers, but a minimum self-contained explanation in the current version of the article must be provided. For instance, in another paper of yours, I read that the images were fetched from the SDSS Sky server using the “cutout” web- service, and the output images were120×120 colorJPEG images. So, does it mean that these are automatically coloured images from the digital survey? Another question is, why use JPEG images and not the original FITS astronomical standards? Aren't you losing a tremendous spatial resolution in this procedure?
Concerning the surveys, why not use the maximum number of SDSS galaxies to construct the statistically complete analysis? I fail to understand why only ~60k galaxies have been used.
A question regarding the method. If instead of binarity (clockwise and counterclockwise) is used, if somehow you could use the angular momentum vector, could the conclusions change? I'm asking this because the anisotropy found seems very small for a significant conclusion, or maybe I did not understand the plots. So, this conclusion, a 3D distribution of the angular momentum vector could wash out a bias of choosing only 1 or -1, i.e. clockwise or counterclockwise results.
Some galaxies in the image examples look like elliptical or lenticulars but still present oscillations. Could this be an artefact from bad co-registering of the bands? Could this be an artefact from poor resolution? How do you get the correct morphological classification for these systems? Have you used SBP profiles?
I was a bit concerned with the reliability of the code Ganalyzer. Has it been tested against fictional datasets? I also have some extra Ideas for testing the reliability of the Ganalyzer code. BTW, is the code publicly available? If so, it could be interesting to have a link/webpage pointing to it.
1. An idea is to use Integral Field Spectroscopy (IFS) of nearby galaxies combined with photometric bands and check whether the spin is correctly recovered. From the 2D morphological maps, one can have sufficient resolution in these surveys to make the test.
2. Following part 1 of this exercise, Use IFS data to degrade the data to lower and lower resolutions and still see if you can get the same results with Ganalyzer. Is there a limit? Has this kind of test been done?
Concerning the article's content, how does this article differ from previous ones you have published? (113, 116, 120, 121... ???) I found it not very clear what the main differences from the analyses presented here. Additionally, I find too many self-citations in some parts of the text and wonder if other studies could be contrasted or if, indeed, you are the only one doing that in the field. As another point in your paper, studying the galaxy spin direction distribution in HST and SDSS shows similar large-scale asymmetry (Figure 3 of that paper) to me, which is very different from figure 6 of the current paper.
Now, some general comments regarding the text:
Abstract: "All five sky surveys show similar profiles, exhibiting dipole axes
well within 1s error from each other." I found this phrase a bit misleading because the 1 sigma error range is really large, so basically is not so difficult to have the axes well within. However, when looking at the probability of a dipole axis in galaxy spin directions from different (alpha, delta) combinations the maps look similar except for the DES survey. Maybe I failed this part of the explanation in the text, could you clarify a bit further and also expand this part of the explanation?
1. Introduction
page 1
"Black holes spin [89–94], and 32 their spin is inherited from the spin of the star from which the black hole was created [92]." But not all black holes are known how to form. For instance, the origin of supermassive black holes in the Universe is not yet understood. Why 'stars' are assumed as the seed? What is the initial “seed” of supermassive black holes formed in the early Universe? How massive they were? In what types of galaxies do they form?
page 2
"A spiral galaxy is a unique astrophysical object in the sense that its visual appearance depends on the location of the observer." I failed to understand this sentence. Is the location (distance, the viewing angle or both?) that changes most the visual appearance of spirals?
page 2
"2.1. Automatic annotation of galaxies by their spin direction" - Comment/Question
I understand the problems exposed by pattern recognition or deep learning systems. Still, model-driven approaches are subject to the incompleteness of the ingredients in the models, which can cause biases.
page 3
"Additionally, training such pattern recognition systems requires a manually annotated training set." Just a curiosity question: even for unsupervised learning? I understand that this is the case for supervised learning because one needs labels for the training set, but could we use an unsupervised learning algorithm that learns patterns from untagged data?
"Ganalyzer transforms each galaxy image into its radial intensity plot transformation. The radial intensity plot of an image is a 35 x 360 image..." I fail to understand why 35 x 360 and not 36 x 360 or 38 x 360, ... I assume that the first number are different distances from the centre? So, why 35? I found in one paper that radial distances go from 0.4 to 0.75, but why these values are chosen?
For a non-symmetric system, e.g. irregular galaxy, how the algorithm selects the centre so that it does not introduce bias? BTW, irregular galaxies are not mentioned in the text. Would this type of morphology introduce biases in the analysis if wrongly classified as a spiral?
"Obviously, not all galaxies are spiral galaxies, and not all spiral galaxies have a clear identifiable spin direction. Therefore, most of the galaxies cannot be used for the analysis due to the inability to identify the direction in which they spin. For that reason, only galaxies that have 30 or more identified peaks in the radial intensity plot are used in the analysis." My question regarding this part is the following if the majority of galaxies do have not a clear identifiable spin direction, how can you be sure this cannot introduce biases in the analysis? Also, if only galaxies with more than 30 peaks are chosen, can spin direction be dependent on the number of peaks or degree of spirality?
page 3-4
When it comes to the datasets, it is important to understand the photometric bands used. Combined or individual (see my comments above). Also, nowhere clearly is stated the spatial resolution of images from each dataset. I also wonder how this would impact the results. Maybe it is shown somewhere but could be expanded and clarified in the text.
Figure 1 could be slightly bigger, and the caption should better explain the radial intensity plots. What are the Y and X directions in the right part of the diagrams?
"The size of each image is 256 256, and retrieved in the JPEG format." Why use JPEG format? Are these colour images? I've already commented on my concerns in the first part of the report above.
page 5
"the annotation, full consistency was ensured by using just one computer system with a single processor." Does the method depend on the computer specifications? Does it pass the reproducibility test? Do I obtain distinct results if I run the same galaxy in different environments?
page 6
2.4. SDSS data
"The SDSS dataset contains 63,693 galaxies [119,121]." Please, rephrase it. The chosen subsample of SDSS galaxies has 63693 galaxies
page 7
"While the automatic annotation is fully symmetric, it also leads to the sacrifice of some galaxies that their spin direction cannot be determined." Still don't understand why. Also, what is the typical fraction lost by non-determination of the spin direction algorithm? I think this should be stated for each sample. Maybe dependent on the dataset.
page 8
"While this separation of this sky does not have cosmological meaning, ..." Explain a little bit better why it has no cosmological meaning. Is there any segmentation that would have?
page 10
"SDSS galaxies were selected such that their redshift distribution was similar to the distribution of the subset of DESI Legacy" I don't understand why. SDSS is unique because there are several millions of galaxies with redshift. Why not use all datasets? Why select only a subset? Why select a subset matching similar distribution as DESI Legacy?
page 11
In Figure 6 - Does it make sense to produce a plot combining all datasets together and deriving the probability of a dipole axis in galaxy spin directions? If not, why? What kind of biases can arise?
page 12
"Machine earning and pattern recognition systems are often highly complex and unintuitive." Substitute earning by learning.
Non-intuitive to development is one thing, but if the algorithms work, this is no reason to attack them. Also, complexity is not a valid argument if, again, it works! I suggest rephrasing to supervised learning as I'm not sure the arguments might still be valid for unsupervised learning. Maybe you could comment on that to clarify.
page 13
"The retrieval of the image data and the annotation of the galaxy images were done on the same computer system, in order to avoid any kind of differences between the computers."
Well, you could test using different computers and see whether the results change or are consistent. Also, why it should change between different computers? Can you elaborate a bit more? Shouldn't statistically you obtain the same results?
page 14
Maybe I'm not getting this part right. Same fields can preferentially probe filaments, no? If there is a preference for galaxy orientation in filaments, then analysing the same field would impact the results, no? Shouldn't we combine the maximum number of different fields to obtain a non-biased result?
Although galaxies do not form from the same material/cloud, they seem to not be randomly distributed in space. On the contrary, locally, they seem to follow filaments. So, proto-galaxies are gravitationally connected in small regions of the Universe. This is presumably linked to the underlying distribution of dark matter throughout the Cosmos. The question is how matter tends to move in these filaments. A preferential spin axis can be induced if there is a tendency to move to the highest densities following a corkscrew motion.
"While sich pipelines are difficult to fully ensure are symmetric" - Change sich by such. This phrase is a bit odd.
page 15
"theoretically that machine learning system was expected to be symmetric between clockwise and counterclockwise galaxies. Still, for showing randomness in the annotation of the data, removal of attributes was required. " And... Was not it done? I don't understand this last part. Be clearer.
page 16
"The tables show that the statistical significance of the distribution becomes insignificant when the redshift ranges are lower." Why does it happen? Can you elaborate a bit more on that? If you have slices at higher redshifts, what would happen?
"it is clear that the distribution of the galaxy spin directions in that dataset is not random."
I'm wondering about the effect of spin direction as a function of redshift. Limiting to lower redshift, we don't see anything, but not limited we do see. Not limiting, we will preferentially get bigger, brighter, massive spiral galaxies as the redshift increases, no? Because the code will miss the spin rotation of smaller galaxies (not to mention the morphology classification) due to loss of resolution, correct? What would be this effect in the analysis? Also, wouldn't massive/more giant galaxies be preferentially in denser environments? Another aspect is that galaxies at higher redshifts tend to be fuzzy/irregular; how do determine their spin direction in those cases? Is there an impact on the analysis?
page 17
"The provides a probability" - This?
page 18
"spiral galaxies to examine the nature of the distribution of spin directions of spiral galaxies in the sense of the large-scale structure."
I find it very interesting but hard to understand the morphological classification of spiral systems as a function of redshift because of the loss of resolution in the surveys. Also, galaxies at higher redshifts are probably more turbulent, so the morphological classification tends to become even more challenging. This may prevent the determination of spin directions as we go to higher redshifts. How can this impact the results of this research?
"The observation that galaxies in opposite lines of sight show opposite spin directions also agrees with cosmology driven by longitudinal gravitational waves [204]," But these theories predict by how much the asymmetry? Because the asymmetry seen is not very high.
page 19
"But the observation by five different sky surveys, all showing similar profiles," I don't see similar profiles in all of them. I fail to see the same structure for DES dipole axis maps
Author Response
I would like to thank you for the opportunity to revise this manuscript. The article is in good shape and a very interesting topic transversal to extragalactic astronomy and the Universe's large-scale structure. I understand the humungous work done here and want to congratulate you on this effort. Several possible biases were analysed in the text. First, I would like to recommend modifying the text and clarifying some parts, especially for people not from the area.
--Author response: I would like to thank you for the time you put in reviewing this not-so-short manuscript, the insightful comments, and the very kind words. I am grateful for that. I was given 10 days to address the comments, but I believe that as been done, and the comments have been addressed. Many changes have been made according to the comments. The changes are highlighted in the manuscript in bold font. The responses to the comments and description of changes made to the manuscript are made below each comment.
My comments are as follows:
I understand that the focus of the article is on the large-scale structure of the Universe, but I missed a more general introduction with citations and a connection with galaxy formation and evolution. For instance, how the spin of galaxies is an essential quantity to understanding the evolution of galaxies over cosmic time. Also, it would be beneficial for the article to know whether it is expected from galaxy formation and evolution scenarios that galaxies on large scales are organised within a network of filaments and walls and if it is expected that this large-scale structure produces (locally) anisotropy or not. For example, if this is true, I missed a discussion regarding how small portions of the sky could introduce biases in the analysis.
--Author response: Thank you for the comment. There are several parts to the comment, and I address them here. First of all, it is definitely correct that multiple studies in the past decade showed a correlation between spin alignment and the large-scale structure. A few papers were mentioned in the original version of the manuscript, but many more (23 references) have been added to the Introduction section of the revised version. These papers include observational studies (15 references) and also numerical simulations. Another thing that was added is the “tidal torque theory”, which is currently the “mainstream” explanation to the initial galaxy angular momentum. There is currently a certain agreement on tidal torque theory, and that theory also makes a direct link between the galaxy angular momentum and the cosmic initial conditions. References to these studies have been now added, starting the work of Fred Hoyle from 1949. Additionally, numerical simulations also showed alignment in angular momentum in the context of the large-scale structure. All that has also been added to the revised version of the manuscript. While this study is obviously several orders of magnitude larger in terms of the number of galaxies, the link between alignment in angular momentum and the large-scale structure has been studied in the past, and these papers and discussions have been added to the revised version of the manuscript. Another explanation to initial galaxy spin is galaxy mergers, and there is some more recent work, namely the “spin-flip” phenomenon, showing that spin can be initiated through the initial cosmic conditions, and flip through mergers in higher mass environments. That has also been added with the references to these papers. The Introduction section was expanded substantially to discuss all the topics mentioned above.
The other part of the comment is obviously related to the first part, and that is whether asymmetry in a small part of the sky can lead to a detection of a dipole. A new sub-section (4.7) has been added to discuss that. In summary, the answer to that is yes, and an example to that has been shown in (Shamir, 2021, Particle), where an artificial dataset that was created with high asymmetry in one part of the sky showed fitness to a dipole axis, even when the distribution in the rest of the sky was random. That is probably not the case here. Data from Table 4, and especially Figure 5, show that the asymmetry changes in different parts of the sky, so that it is not just one part of the sky that drives the statistical signal of the dipole axis. Table 8 shows that the axes are consistent across different sky surveys with different footprints, including footprints that do not overlap such as DES and SDSS. The asymmetry in the COSMOS field also shows that, because that field was obviously not chosen for asymmetry in galaxy spin, but it still shows that asymmetry in all telescopes. All that show that it is unlikely that asymmetry in just one part of the sky drives the entire analysis. A new subsection 4.7 has been added to describe that entire discussion.
Indeed, when revising the tidal torque theory (Hoyle 1949; Peebles 1969; Doroshkevich 1970; White 1984; Catelan & Theuns 1996; Lee & Pen 2000, 2001, and Schäfer 2009, for a review), it has been proposed that galaxies acquire angular momentum via the tidal field from surrounding galaxies. In the Lambda Cold Dark Matter (ΛCDM) paradigm, galaxies are first formed in a dark matter halo and evolve through continuous merging and interaction with nearby galaxies while gaining or losing mass and angular momentum. I wonder if, after establishing and correcting for possible biases, we can find the winding sense of the galaxies to be consistent or not with statistical isotropy.
--Author response: That’s a very good point. The alignment in spin directions observed in this paper covers a much larger part of the sky compared to the previous studies. But since galaxy angular momentum is related to the initial conditions, it can also be related to tidal torque theory. In that case, the link with initial conditions goes far beyond the scale of a filament, supercluster, or wall. But because the previous studies were far smaller (in footprint and number of galaxies), they could not have observed that.
According to (Cadiou et al., 2021), tidal torque theory and spin through galaxy mergers are competing explanations. While in tidal torque theory the initial spin is directly related to the cosmic initial conditions, spin through galaxy mergers is not. If galaxy mergers were the only agent that causes galaxy spin, the distribution of galaxy spin directions would have been expected to be completely random. As shown in this paper (and also in many previous smaller-scale papers mentioned above), that is not the case.
According to previous studies such as (Welker et al., 2020), the initial angular momentum of galaxies is related to the initial conditions, while galaxy mergers also make a later impact. Numerical simulations also showed the correlation between spin alignment and the mass (the mass grows with mergers).
As for changes made to the manuscript, the references mentioned in the comment have been added, in addition to several other related references. That has been done in the Introduction section. Also, the discussion about the tidal torque theory, spin-flip, and the observations and simulations supporting it have been added to the Introduction section. References to these papers have also been added. Also, a discussion of four-paragraphs was added to the Discussion section, that includes the spin-flip phenomenon, and the contention that spin directions are related to the initial conditions.
I've also seen that previous studies have shown that the spin of galaxies is correlated with environments, axes of large-scale structure, 2D correlation, and clustering (Porciani et al. 2002; Davis & Natarajan 2009; Tempel et al. 2013; Casuso & Beckman 2015; Codis et al. 2015; Lee et al. 2018b). Is there a possibility for these environmental biases in the current analysis? My point here is for the reader to understand a little bit better what is expected for galaxy spinning locally, connecting with galaxy formation/evolution and globally in terms of cosmological theories. This should be approached in the introduction.
--Author response: Yes. That has been added to the Introduction, and also to the Discussion section, including the references. The addition to the Discussion section was important to emphasize that the study shows evidence of spin alignment in scales larger than the scale of clusters, walls, or filaments. Previous studies showed alignment within certain elements of the large-scale structure, while this study shows that the scale of the alignment is even larger than what has been shown in the past.
Concerning the code Ganalyzer, I had difficulty understanding how the code works beneath the general mathematical explanation. I searched in the original Shamir (2011) paper and other articles, but still, there are some missing gaps. For instance, it seems that the images used as input for the code come from the automatic pipeline of the digital surveys, i.e. the coloured ones by combining different filters. For example, SDSS has created colour images by taking the middle three bands (g, r, i) and assigning those to the blue, green and red channels. And then, you have downloaded these coloured images as JPEG formats? Is this correct? It seems that this is the approach for others, such as in DES, for example. My question is: are you taking the artificially/pre-processed colour images from the digital surveys in JPEG format? I found this part somewhat awkward and confusing in the paper and propose that this part is clarified with more details. Additionally, If this is right, the approach presented in the paper could be inappropriate. It is well known that careful post-processing of imaging data is needed for higher accuracy with surface photometry techniques. For instance, Imaging data from surveys are usually provided in reduced form (e.g., 'u', 'g', 'r', 'i' and 'z' SDSS) after several pre-reduction steps:
1. Removal of instrumental/detector effects (bias, dark and flat-field, saturation, bad pixels);
2. Correction for cosmic ray hits;
3. Computation of calibration constants;
4. Approximate subtraction of the sky background through a low-degree polynomial;
5. Co-registration of images in some or all bands observed
However, the quality of reduction steps [4] & [5] is generally not sufficient for meticulous photometry. This is especially true for surface photometry profile (SBP) analysis and the determination of colour maps. For example, a minor error of typically less than 20% of the sky photon noise in the removal of the local sky background can result in a down-bending or flattening of SBPs in their outer parts, leading then to significant errors in the determination of e.g., the structural parameters of the disk component. Likewise, minor errors in the co-registration of images in two different bands can propagate into strong artefacts in colour maps computed by the division of these images. If JPEG images are used, then these are coloured images. As far as I understand, Ganalyzer provides radial intensity plots, nearly similar technique as SBPs, though not azimuthally averaged. If you are doing your own background removal and co-registration of images or even using one band at each time please clarify. Otherwise, I seriously wonder if your analyses affect the final conclusions, for instance, with the introduction of peaks/oscillations due to misalignments.
--Author response: Thank you for the comment. Section 2 has been revised significantly to make it clear how the images are downloaded and analyzed. The image format is indeed in JPEG format. Downloading the DES data alone took more than six months to complete in the JPG format. Given that in this case the FITS files are about 15 times larger than the JPEG files of the same image, and that each FITS file provides a single channel, it would have taken more than 20 years just to download the images of the ~18 million DES galaxies. That has been added to the beginning of Section 2 (in bold font). That shows that using FITS as the file format is not practical given the size of the data. Only the HST galaxies are in FITS format, because the number of galaxies in HST is much smaller.
The JPEG images are converted to grayscale, and the directions of the arms are determined by differences in the pixel intensities (identifying the brightest pixels at the same radial distance from the center of the galaxy). The JPEG format has the visual information in it, and allows to identify the spin direction. Using the JPEG format might lead to some galaxies that their spin direction cannot be identified. That, however, is expected to be have similar impact on images of galaxies that spin clockwise and images of galaxies that spin counterclockwise. That discussion is now summarized in the second and third paragraphs of Section 2 (the new paragraphs are highlighted in bold font).
Connected with the previous paragraph, I also profoundly missed a detailed description of the datasets and what is used as input. The photometric bands or the combination of them used for each instrument. For instance, SDSS has five bands, is this analysis used in each one of the bands or in the colour image? I found the description of the datasets extremely poor and missing details. I understand that these may have been discussed in previous papers, but a minimum self-contained explanation in the current version of the article must be provided. For instance, in another paper of yours, I read that the images were fetched from the SDSS Sky server using the “cutout” web- service, and the output images were120×120 colorJPEG images. So, does it mean that these are automatically coloured images from the digital survey? Another question is, why use JPEG images and not the original FITS astronomical standards? Aren't you losing a tremendous spatial resolution in this procedure?
--Author response: A lot of the information has been addressed in the reply to the previous comment. Using the FITS format is not practical due to the gigantic amounts of data that need to be downloaded. Downloading the DES data took over six months, and with the FITS format it will take an order-of-magnitude more time, making it impractical. In all cases the cutout service was used, and that was added to the paper (Section 2). The color images were converted to grayscale, and therefore each image was analyzed once, rather than several times, each time for a different color. That has also been added to the description. That was done for all datasets, so the description is in the beginning of Section 2, rather than for each dataset separately.
Concerning the surveys, why not use the maximum number of SDSS galaxies to construct the statistically complete analysis? I fail to understand why only ~60k galaxies have been used.
--Author response: The SDSS dataset was taken from a previous paper. The initial dataset was the entire set of galaxies in DR14 that had spectra, and Petrosian radius larger than 5.5”. These galaxies were annotated, and about 64K of them had identified spin directions. That has been added to the revised Section 2.4. Because the requirement for the galaxies to have spectra, the dataset was smaller than what it could have been without that requirement. But the spectra were very useful for other purposes such as identifying the increasing asymmetry with the redshift (which have also been added to the paper in Table 9). In general, preparing each of these datasets requires very substantial time to download and analyze the data.
A question regarding the method. If instead of binarity (clockwise and counterclockwise) is used, if somehow you could use the angular momentum vector, could the conclusions change? I'm asking this because the anisotropy found seems very small for a significant conclusion, or maybe I did not understand the plots. So, this conclusion, a 3D distribution of the angular momentum vector could wash out a bias of choosing only 1 or -1, i.e. clockwise or counterclockwise results.
--Author response: That is a good question, but I’m afraid that at this point I don’t have the data to answer it. The magnitude of the asymmetry is indeed small, but because of the large number of galaxies the statistical significance is high. The DES axis is with significance of 3.7 sigma, and even very simple separation of the data to opposite hemispheres shows strong statistical significance. A short note has been added to the conclusion section.
Some galaxies in the image examples look like elliptical or lenticulars but still present oscillations. Could this be an artefact from bad co-registering of the bands? Could this be an artefact from poor resolution? How do you get the correct morphological classification for these systems? Have you used SBP profiles?
--Author response: The image examples are in Figure 1, and all of them are spiral galaxies. In general, the spin direction is determined from the alignment of the peaks. Elliptical galaxies might also have peaks. For instance, an elliptical galaxy that has certain inclination might have some peaks. But the peaks will be aligned in a straight line, and will not show a shift towards a certain direction. An example is Figure 2 in (Shamir, 2011, ApJ). The elliptical galaxies have peeks, but the peeks make a straight line, with no preference to a certain direction. That short description has been added to Section 2.1, with a reference to (Shamir, 2011).
I was a bit concerned with the reliability of the code Ganalyzer. Has it been tested against fictional datasets? I also have some extra Ideas for testing the reliability of the Ganalyzer code. BTW, is the code publicly available? If so, it could be interesting to have a link/webpage pointing to it.
1. An idea is to use Integral Field Spectroscopy (IFS) of nearby galaxies combined with photometric bands and check whether the spin is correctly recovered. From the 2D morphological maps, one can have sufficient resolution in these surveys to make the test.
2. Following part 1 of this exercise, Use IFS data to degrade the data to lower and lower resolutions and still see if you can get the same results with Ganalyzer. Is there a limit? Has this kind of test been done?
--Author response: The code for Ganalyzer was published through the Astrophysics Source Code Library (https://www.ascl.net/1105.011) since 2011, but I have not updated it since then. So far it has been tested against actual image data, but not against synthetic images. Tests have been done on galaxies annotated manually. If a galaxy has leading arm, Ganalyzer will fail to identify its spin direction, and therefore might disagree with IFS. That is expected, so full agreement with IFS is not expected. As discussed in Section 4.6., such galaxies are expected to be distributed evenly between clockwise and counterclockwise galaxies, and are expected to weaken the signal rather than artificially strengthen it. Ganalyzer was tested on smaller galaxies, and when the radius is smaller than 5.5”. it becomes less effective. For that reason all galaxies analyzed by Ganalyzer are limited to radius of 5.5” or larger.
Concerning the article's content, how does this article differ from previous ones you have published? (113, 116, 120, 121... ???) I found it not very clear what the main differences from the analyses presented here. Additionally, I find too many self-citations in some parts of the text and wonder if other studies could be contrasted or if, indeed, you are the only one doing that in the field. As another point in your paper, studying the galaxy spin direction distribution in HST and SDSS shows similar large-scale asymmetry (Figure 3 of that paper) to me, which is very different from figure 6 of the current paper.
--Author response: The main contribution of the paper is the analysis of DES, and the comparison of the results with other sky surveys analyzed previously. DES is the deeper than most other sky survey analyzed for this purpose so far, and it is a well-known survey. The results shown in this paper were presented at the last AAS Meeting last month, but were not published yet. The results of DES and the comparison to the other sky surveys could be interesting to many of Universe readers. I tried to remove as many self-citations as possible. I found four self-citation I could safely remove. Other references cite data or results that are being used or compared to, and these references are therefore needed. I cite all papers on the field I am aware of, including some that show the same conclusions as mine (mentioned in Section 1), but also those with opposite conclusions (discussed in Section 5).
The SDSS chart is indeed different from the original SDSS chart. That is because just a subset of the SDSS dataset was used, such that the redshift distribution was normalized to the redshift distribution of DESI, As the original version of the paper says:
“the SDSS galaxies were selected such that their redshift distribution was similar to the distribution of the subset of DESI Legacy Survey galaxies with redshift obtained through 2dF. That resulted in a dataset of 38,264 galaxies such that the distribution of the redshifts of the galaxies fits the distribution of the redshift of the galaxies in the DESI Legacy Survey.”
So instead of 64K galaxies, 38,264 galaxies were used such that the redshift distribution was similar to the redshift distribution in DESI. As shown in the previous papers (now cited in the paper), the location of the peak changes with the redshift, and therefore normalization of the redshift is expected to lead to a close location of the dipoles, which is indeed what happened. Some changes were made in the text to make it more clear. Perhaps more importantly, a note was added also to the caption of the figure to make it more noticeable for the reader that the SDSS galaxies are a subset of the SDSS dataset used in previous papers.
Now, some general comments regarding the text:
Abstract: "All five sky surveys show similar profiles, exhibiting dipole axes
well within 1s error from each other." I found this phrase a bit misleading because the 1 sigma error range is really large, so basically is not so difficult to have the axes well within. However, when looking at the probability of a dipole axis in galaxy spin directions from different (alpha, delta) combinations the maps look similar except for the DES survey. Maybe I failed this part of the explanation in the text, could you clarify a bit further and also expand this part of the explanation?
--Author response: I agree that sometimes we tend to follow standard statistical practices that do not provide all information. The 1 sigma error is a common statistical practice to compare results of simulations, but it does not show the full story. The abstract was changed so that the angular distance between the axes is now specified. The angular distance of no more than 52 degrees between the axes is based on Table 8, where the largest angular distance is between the axis observed with DES and the axis observed with Pan-STARRS.
The statement about “similar profiles” was also removed. The DES pattern is somewhat different from the others, but given that each sky survey has a different footprint, different limiting magnitude, different number of galaxies, etc, some differences are expected. But while the degree of “similarity” involves some art with science, the locations of the most likely axes can be compared numerically. That has been added to the abstract. DES is the deepest sky survey but also has the smallest footprint, and that could also lead to a certain difference as a large footprint is expected to get better accuracy. That has also been added to the discussion of Figure 6.
- Introduction
page 1
"Black holes spin [89–94], and 32 their spin is inherited from the spin of the star from which the black hole was created [92]." But not all black holes are known how to form. For instance, the origin of supermassive black holes in the Universe is not yet understood. Why 'stars' are assumed as the seed? What is the initial “seed” of supermassive black holes formed in the early Universe? How massive they were? In what types of galaxies do they form?
--Author response: Thank you for the comment. The original version did not make a distinction between a stellar black hole and a supermassive black hole. The introduction section was corrected to “stellar black hole” to avoid the confusion, and a quick note was added that further discussion is available in Discussion section. The discussion section was also changed, and now there is a distinction between stellar black holes and supermassive black holes. As for supermassive black holes, their formation is more debatable than stellar black holes, but several theories have been proposed, including collapse of supermassive stars (Reese, 2006; Shibata et al. 2002) black hole mergers (Pacucci & Loeb, 2020), or “direct collapse” (Habouzit et al., 2016). But regardless of the theory, recent work, namely by Chris Reynolds (2013, 2019), showed that supermassive black holes also spin. All of that discussion has been added with the references to the discussion in Section 6.
page 2
"A spiral galaxy is a unique astrophysical object in the sense that its visual appearance depends on the location of the observer." I failed to understand this sentence. Is the location (distance, the viewing angle or both?) that changes most the visual appearance of spirals?
--Author response: In the revision of the Introduction section that sentence has been removed. The idea was that if you look at a galaxy from its opposite side its spin direction would flip, but that is rather obvious so the presence of the sentence became somewhat confusing. In any case, it did not contribute much to the discussion and was removed in the revised version.
page 2
"2.1. Automatic annotation of galaxies by their spin direction" - Comment/Question
I understand the problems exposed by pattern recognition or deep learning systems. Still, model-driven approaches are subject to the incompleteness of the ingredients in the models, which can cause biases.
--Author response: It is true that model-driven algorithms might also be biased, although the possibility of bias is reduced dramatically when the rules are interpretable and defined (and therefore can be symmetric by design, which is not possible with machine learning/neural networks). They are also much easier to test and analyze compared to machine learning algorithms where the rules are complex and uninterpretable. The majority of that discussion is done in Section 4, but a note has been added to Section 2.1 to indicate that while the possibility of bias is reduced, it is still not 0.0%.
page 3
"Additionally, training such pattern recognition systems requires a manually annotated training set." Just a curiosity question: even for unsupervised learning? I understand that this is the case for supervised learning because one needs labels for the training set, but could we use an unsupervised learning algorithm that learns patterns from untagged data?
--Author response: Unsupervised learning would not require manual annotations, but might have other challenges because these algorithms have substantial noise, and they learn from anything they find to divide the data into classes. For instance, unsupervised learning can divide the galaxies into small galaxies and large galaxies, or to dim galaxies and bright galaxies, and so many other things that the algorithm can pick to create classes. Because it can learn from anything, and combinations of features, it becomes very difficult to control or verify it. While I would not have used machine learning for a task like this, I also understand that it is debatable, and some might argue that the use of unsupervised machine learning can be valid. A short note about it has been added to the section.
"Ganalyzer transforms each galaxy image into its radial intensity plot transformation. The radial intensity plot of an image is a 35 x 360 image..." I fail to understand why 35 x 360 and not 36 x 360 or 38 x 360, ... I assume that the first number are different distances from the centre? So, why 35? I found in one paper that radial distances go from 0.4 to 0.75, but why these values are chosen?
--Author response: As also discussed in (Shamir, 2011, ApJ), several different ranges were used and tested. But close to the center (<0.4) there is not a lot of information, and also in the outskirts of the galaxy (>0.75) the arms start to fade. The range of 0.4-0.75 gives the most informative part of the arm. A note has been added.
For a non-symmetric system, e.g. irregular galaxy, how the algorithm selects the centre so that it does not introduce bias? BTW, irregular galaxies are not mentioned in the text. Would this type of morphology introduce biases in the analysis if wrongly classified as a spiral?
--Author response: In the case of irregular galaxies it is not supposed to identify consistent shift in the position of the arms. Even if the center cannot be determined accurately, it is not expected to show consistent change in the location of the arms. In case arms are identified with consistent shift, it might mean that the galaxy spins. I am not familiar with a problem with irregular galaxies, but since irregular galaxies can come in all shapes it is theoretically possible that irregular galaxies might be identified as if they have a spin direction, although that spin direction is in fact random. Galaxies with clear arms but no clear center are not common (I tested all types of galaxies and I am not familiar with such problem), and in any case because the algorithm is symmetric is expected to be divided evenly between clockwise and counterclockwise galaxies. If such thing happens, it can therefore only reduce the signal but cannot artificially increase it. The is also now discussed in Section 4.1.
"Obviously, not all galaxies are spiral galaxies, and not all spiral galaxies have a clear identifiable spin direction. Therefore, most of the galaxies cannot be used for the analysis due to the inability to identify the direction in which they spin. For that reason, only galaxies that have 30 or more identified peaks in the radial intensity plot are used in the analysis." My question regarding this part is the following if the majority of galaxies do have not a clear identifiable spin direction, how can you be sure this cannot introduce biases in the analysis? Also, if only galaxies with more than 30 peaks are chosen, can spin direction be dependent on the number of peaks or degree of spirality?
--Author response: A new sub-section 4.9 has been added to address that. First of all, regardless how good the algorithm is, some galaxies that have a spin direction will still be ignored from the analysis. For instance, Figure 10 has been added to show some galaxies that according to SDSS are elliptical and do not have an identifiable spin direction, but in HST the same galaxies have very clear spin direction. So an algorithm applied to SDSS might ignore these galaxies, while these galaxies do have a spin. So any algorithm will always ignore some galaxies that do spin. The question is how we work with that. To that there are two answers:
1) There is no apparent reason for that to happen. The algorithm is symmetric, so the number of clockwise galaxies being rejected is expected to be the same (within statistical error) as the number of counterclockwise galaxies. I understand that it was a serious problem with the Galaxy Zoo data, and might also be a problem with machine learning. But the human brain is biased, and machine learning may or may not be biased so it is difficult to control.
2) Even if such bias existed, it would have been expected to be consistent throughout the entire sky. A simple mathematical analysis has been added to the new sub-section 4.9.
page 3-4
When it comes to the datasets, it is important to understand the photometric bands used. Combined or individual (see my comments above). Also, nowhere clearly is stated the spatial resolution of images from each dataset. I also wonder how this would impact the results. Maybe it is shown somewhere but could be expanded and clarified in the text.
--Author response: The information has been added to Section 2. The images used the cutout services of the sky surveys, which use designed algorithm to create the color images. The revised version now has the references to these algorithms, or the short description in the text when these algorithms are not available. In SDSS, the algorithm is described in (Lupton et al., 2014), and the reference has been added.
Pan-STARRS is more simple, where the RGB images are constructed such that the R value is the y band, the G value is the i band, and the B value is the g band.
DECam (DES and DESI) uses the calibrated z, r, and g bands of DECam as the R, G, and B values of the RGB pixels.
The scale is determined by the radius of the galaxy, such that the galaxy fits the frame. That has been added to Section 2. That is a generic way to ensure that the image fits the frame, so changing the scale will not affect the analysis as the scale is determined by the size of the galaxy. In any case, testing other scales will require to re-download the data, which is a process that takes six months or more.
Figure 1 could be slightly bigger, and the caption should better explain the radial intensity plots. What are the Y and X directions in the right part of the diagrams?
--Author response: Thank you for the suggestion. That has been done. The idea was to fit the image into a single column in the final version, but in that case the figure is indeed more difficult to view. The figure is now larger, and the caption is more detailed.
"The size of each image is 256 256, and retrieved in the JPEG format." Why use JPEG format? Are these colour images? I've already commented on my concerns in the first part of the report above.
--Author response: Using the FITS format is impractical due to the huge size of the data being downloaded. What took more than 6 months to download in JPEG, will take years to download with FITS. The data is gigantic. That has been addressed in several previous comments, and added to the manuscript.
page 5
"the annotation, full consistency was ensured by using just one computer system with a single processor." Does the method depend on the computer specifications? Does it pass the reproducibility test? Do I obtain distinct results if I run the same galaxy in different environments?
--Author response: It is very difficult to think of a reason for differences between different computers. This was done as a matter of extra precaution. For instance, if one hemisphere was downloaded by one computer and another hemisphere by another computer, we might think that a difference between them might have been due to different computers. The reason it was done was just to completely eliminate that possibility so it should not be a subject of matter. A short note has been added to Section 2.
page 6
2.4. SDSS data
"The SDSS dataset contains 63,693 galaxies [119,121]." Please, rephrase it. The chosen subsample of SDSS galaxies has 63693 galaxies
--Author response: That was changed as suggested, although, as explained in the reply to a previous comment, these were all galaxies that could be used because only large, bright galaxies with spectra were used, which reduced the number of available galaxies.
page 7
"While the automatic annotation is fully symmetric, it also leads to the sacrifice of some galaxies that their spin direction cannot be determined." Still don't understand why. Also, what is the typical fraction lost by non-determination of the spin direction algorithm? I think this should be stated for each sample. Maybe dependent on the dataset.
--Author response: That sentence was indeed confusing, and was removed. But as for the question, most galaxies are either not spiral, have arms that are too faint, or have arms with no clear spin direction. In DES, just 4% of the initial set of objects are galaxies with identifiable spin directions.
page 8
"While this separation of this sky does not have cosmological meaning, ..." Explain a little bit better why it has no cosmological meaning. Is there any segmentation that would have?
--Author response: That is indeed a little confusing. The idea was to separate the sly into two arbitrary hemispheres, not driven by any previous knowledge. The celestial coordinates have a meaning compared to Earth, but assuming the Copernican principle they have no cosmological meaning. That is, the sky of the Western hemisphere should be the same as the sky in the Eastern hemispheres, and there should be no difference between the Universe in the Eastern hemisphere and the Universe observed in the Western hemisphere. The simple separation is done for the sake of very simple statistical analysis that avoid any kind of “sophisticated” statistics. A clarification has been added to Section 3.
page 10
"SDSS galaxies were selected such that their redshift distribution was similar to the distribution of the subset of DESI Legacy" I don't understand why. SDSS is unique because there are several millions of galaxies with redshift. Why not use all datasets? Why select only a subset? Why select a subset matching similar distribution as DESI Legacy?
--Author response: That was explained in the replies to one of the previous comments, and the changes have been made to the manuscript. SDSS DR14 indeed has some 3M galaxies with spectra, but the vast majority of them are elliptical galaxies or galaxies that are small and faint to identify their spin direction. That explanation was added to Section 2. The selection of the subset that has similar redshift distribution is because previous results showed that the location of the axis changes with the redshift. Therefore, two dataset with similar redshift distribution are expected to show similar location of the dipole axis. That has been added to Section 3 in the manuscript that discusses Figure 6, and also the caption of the Figure.
page 11
In Figure 6 - Does it make sense to produce a plot combining all datasets together and deriving the probability of a dipole axis in galaxy spin directions? If not, why? What kind of biases can arise?
--Author response: In my opinion it does make sense. Doing that takes a long time, and surely cannot be completed within the 10 days I was given to revise the manuscript. Fortunately, I have already done that before, and the Figure of using nearly 1M galaxies when combining telescopes has been added as Figure 8 in Section 3.
page 12
"Machine earning and pattern recognition systems are often highly complex and unintuitive." Substitute earning by learning.
--Author response: Corrected. Thank you.
Non-intuitive to development is one thing, but if the algorithms work, this is no reason to attack them. Also, complexity is not a valid argument if, again, it works! I suggest rephrasing to supervised learning as I'm not sure the arguments might still be valid for unsupervised learning. Maybe you could comment on that to clarify.
--Author response: The problem with these algorithms is that they learn from the data, and that makes it very difficult to control for the bias. We made thorough analysis of that in (Dhar & Shamir, 2022). I changed the sentence to start with “Supervised machine learning”, although I am not at all convinced that unsupervised machine learning can solve these problems or make it even more difficult to identify biases. I do believe that avoiding the use of machine learning is a substantial advantage.
page 13
"The retrieval of the image data and the annotation of the galaxy images were done on the same computer system, in order to avoid any kind of differences between the computers."
Well, you could test using different computers and see whether the results change or are consistent. Also, why it should change between different computers? Can you elaborate a bit more? Shouldn't statistically you obtain the same results?
--Author response: This comment is closely related to a previous comment. A computer program runs the same on all computers. That’s obvious. But when looking for unlikely reasons for something to happen, using different machines can be one such unlikely reason. Using the same machine just helps to avoid the need to discuss and analyze possible differences between machines (differences that are not supposed to exist anyway). That is now explained in the paper next to that comment.
page 14
Maybe I'm not getting this part right. Same fields can preferentially probe filaments, no? If there is a preference for galaxy orientation in filaments, then analysing the same field would impact the results, no? Shouldn't we combine the maximum number of different fields to obtain a non-biased result?
--Author response: The word “field” is used here in its most general form. In the revised version it is changed to “part of the sky” to avoid confusion. The parts of the sky used here are far larger than any filament or wall. That has been added to the Discussion section.
Although galaxies do not form from the same material/cloud, they seem to not be randomly distributed in space. On the contrary, locally, they seem to follow filaments. So, proto-galaxies are gravitationally connected in small regions of the Universe. This is presumably linked to the underlying distribution of dark matter throughout the Cosmos. The question is how matter tends to move in these filaments. A preferential spin axis can be induced if there is a tendency to move to the highest densities following a corkscrew motion.
--Author response: Thank you. That has been added to the Discussion section.
"While sich pipelines are difficult to fully ensure are symmetric" - Change sich by such. This phrase is a bit odd.
--Author response: Thank you for the comment. I agree that the paragraph was not written well. The entire paragraph has been rephrased.
page 15
"theoretically that machine learning system was expected to be symmetric between clockwise and counterclockwise galaxies. Still, for showing randomness in the annotation of the data, removal of attributes was required. " And... Was not it done? I don't understand this last part. Be clearer.
--Author response. The description in the original version was not clear, and was therefore changed. That paper used SDSS galaxies and showed asymmetry in the same direction as shown with SDSS galaxies in this paper. Only when they removed specifically all the attributes that were able to identify the spin direction of the galaxies, they observed random distribution. That is completely expected after removing specifically all attributes that can identify, even weakly, the spin direction of the galaxies. That explanation has been completely revised, and it is believed to be clearer. When not removing these selected attributes, the distribution of spin directions in not random.
page 16
"The tables show that the statistical significance of the distribution becomes insignificant when the redshift ranges are lower." Why does it happen? Can you elaborate a bit more on that? If you have slices at higher redshifts, what would happen?
--Author response: That has been done in previous papers. The replies to the next comments have the details to how the manuscript was changed. A new sub-section (Section 4.8) has been added to discuss that, but the details are in that section and the replies to the next comments. One table was taken from a previous paper to show the change in asymmetry with the redshift, but there is much more information in the cited paper.
Why does it happen is a question. As explained in the reply to the next comment, it is probably not the size, and that is based on experiments. It might be an indication that the spin directions is driven by the cosmic initial conditions, and becomes more stochastic in time, maybe due to mergers. But that is still something that needs to be studied further with galaxies with higher redshifts than what the current sky surveys can provide. All that has been added to the new Section 4.8.
"it is clear that the distribution of the galaxy spin directions in that dataset is not random."
I'm wondering about the effect of spin direction as a function of redshift. Limiting to lower redshift, we don't see anything, but not limited we do see. Not limiting, we will preferentially get bigger, brighter, massive spiral galaxies as the redshift increases, no? Because the code will miss the spin rotation of smaller galaxies (not to mention the morphology classification) due to loss of resolution, correct? What would be this effect in the analysis? Also, wouldn't massive/more giant galaxies be preferentially in denser environments? Another aspect is that galaxies at higher redshifts tend to be fuzzy/irregular; how do determine their spin direction in those cases? Is there an impact on the analysis?
--Author response: That is obviously a good point. It is definitely correct that the correlation with the redshift is in fact a correlation with the size. That was studied back in 2020, with an experiment that focused on just one (most populated) part of the sky, and changing the limit of the size of the galaxy. The results showed no significant correlation between the magnitude of the asymmetry and the size. According to the data, the correlation is with the redshift, and not necessarily due to the change in the size (and consequently mass) of the galaxies. That observation is also aligned with the contention that spin through galaxy mergers is not expected to be less stochastic (Cadiou et al., 2021). That has been added with a reference to the paper in which the experiment was made to the new subsection 4.8 that discusses the redshift. Also, please note the reply to the next comment, which is also related to the redshift and its impact on the analysis.
page 17
"The provides a probability" - This?
--Author response: Yes. Thank you. That has been corrected.
page 18
"spiral galaxies to examine the nature of the distribution of spin directions of spiral galaxies in the sense of the large-scale structure."
--Author response: The sentence has been revised.
I find it very interesting but hard to understand the morphological classification of spiral systems as a function of redshift because of the loss of resolution in the surveys. Also, galaxies at higher redshifts are probably more turbulent, so the morphological classification tends to become even more challenging. This may prevent the determination of spin directions as we go to higher redshifts. How can this impact the results of this research?
--Author response: A new section 4.8 has been added to discuss it. The effect of the redshift was analyzed with experimental results in a previous paper. A new sub-section 4.8 has been added to briefly discuss the effect of the redshift, with reference to the paper in which it is discussed. Because the galaxies are limited by their size and magnitude, the effect of the redshift is highly limited, as was shown in previous work. That is explained with references to relevant previous work in the new sub-section 4.8.
"The observation that galaxies in opposite lines of sight show opposite spin directions also agrees with cosmology driven by longitudinal gravitational waves [204]," But these theories predict by how much the asymmetry? Because the asymmetry seen is not very high.
--Author response: Personally, that theory is not my first choice as the explanation to the observation. Still, I prefer to include all theories, including those that I personally find less appealing, and perhaps there are things that are incomplete about that theory that will be completed in the future. I added a comment to that paragraph that the observed asymmetry is too mild to support a separation of the entire galaxy population into pairs of antipode galaxies.
page 19
"But the observation by five different sky surveys, all showing similar profiles," I don't see similar profiles in all of them. I fail to see the same structure for DES dipole axis maps
--Author response: These studies compare different telescopes with different galaxies, different limiting magnitudes, different footprints, different size, photometric pipeline, etc. Given that, these profiles are more-or-less as close as they can get. In the revised version, the text was changed into “all showing a statistically significant dipole axis within 1$\sigma$ error from each other”. That sentence is better defined than the previous version, that leaves the word “similar” to the imagination of the reader. The dipole axes are all close across datasets.
Round 2
Reviewer 2 Report
Thank you for the revised manuscript and all the explanations provided. I also would like to congratulate you on the work done. The work is indeed humungous and could not be done with FITS files. Additionally, as I said in my previous report, the result is fascinating and transversal to extragalactic (galaxy formation/evolution) and Universe's large-scale structure.
All my inquiries/concerns were addressed and implemented in the new version of the article. In addition, the author added several extra references to the text. So, I would like to recommend the manuscript for publication.
Thanks for all the effort you made in clarifying those points.